

# Changes in clouds and thermodynamics under solar geoengineering and implications for required solar reduction

Rick D. Russotto[1] and Thomas P. Ackerman[1,2]

[1]Department of Atmospheric Sciences, University of Washington, Seattle, Washington, USA
[2]Joint Institute for the Study of the Atmosphere and Ocean, University of Washington, Seattle, Washington, USA

*Correspondence to:* Rick D. Russotto (russotto@uw.edu)

**Abstract.**

The amount of solar constant reduction required to offset the global warming from an increase in atmospheric $CO_2$ concentration is an interesting question with implications for assessing the feasibility of solar geoengineering scenarios and for improving our theoretical understanding of Earth's climate response to greenhouse gas and solar forcings. This study investigates this question by analyzing the results of 11 coupled atmosphere-ocean global climate models running Experiment G1 of the Geoengineering Model Intercomparison Project, in which $CO_2$ concentrations are abruptly quadrupled and the solar constant is simultaneously reduced by an amount tuned to maintain top of atmosphere energy balance and preindustrial global mean temperature. The required solar constant reduction in G1 is between 3.2% and 5.0%, depending on the model, and is uncorrelated with the models' equilibrium climate sensitivity, while a formula from the experiment specifications based on the models' effective $CO_2$ forcing and planetary albedo is well-correlated with but consistently underpredicts the required solar reduction. We propose an alternative theory for the required solar reduction based on $CO_2$ instantaneous forcing and the sum of radiative adjustments to the combined $CO_2$ and solar forcings. We quantify these radiative adjustments in G1 using established methods and explore changes in atmospheric temperature, humidity and cloud fraction in order to understand the causes of these radiative adjustments.

The zonal mean temperature response in G1 exhibits cooling in the tropics and warming in high latitudes at the surface; greater cooling in the upper troposphere at all latitudes; and stratospheric cooling which is mainly due to the $CO_2$ increase. Tropospheric specific humidity decreases due to the temperature decrease, while stratospheric humidity may increase or decrease depending on the model's temperature change in the tropical tropopause layer. Low cloud fraction decreases in all models in G1, an effect that is robust and widespread across ocean and vegetated land areas. We attribute this to a reduction in boundary layer inversion strength over the ocean, and a reduction in the release of water from plants due to the increased $CO_2$. High cloud fraction increases in the global mean in most models. The low cloud fraction reduction and atmospheric temperature decrease have strong warming effects on the planet, due to reduced reflection of shortwave radiation and reduced emission of longwave radiation, respectively. About 50% to 75% of the temperature effect is caused by the stratospheric cooling, while the reduction in atmospheric humidity results in increased outgoing longwave radiation that roughly offsets the tropospheric temperature effect. Taken together, the sum of the diagnosed radiative adjustments and the $CO_2$ instantaneous forcing predicts the required solar forcing in G1 to within about 6%. The cloud fraction response to the G1 experiment raises interesting ques-





tions about cloud rapid adjustments and feedbacks under solar versus greenhouse forcings, which would be best explored in a model intercomparison framework with a solar-forcing-only experiment.

# 1 Introduction

In light of the warming of Earth in response to anthropogenic greenhouse gas emissions (IPCC, 2013), and continued lack of
progress in curbing those emissions (World Meteorological Organization, 2017), some (e.g. Crutzen, 2006) have argued for serious consideration of solar geoengineering, or reflecting sunlight to artificially cool the Earth, as a means of reducing harms from climate change. The Geoengineering Model Intercomparison Project (GeoMIP; Kravitz et al. (2011b)) was created to study the climate impacts of solar geoengineering schemes. GeoMIP consists of a set of standardized experiments for global climate models (GCMs) that include both an increase in $CO_2$ and some compensating effect, such as a reduction in the solar
constant or an increase in stratospheric aerosol concentration. In Experiment G1, the simplest of the GeoMIP experiments, the $CO_2$ concentration is abruptly quadrupled relative to preindustrial levels, as in the abrupt4xCO2 experiment from the Coupled Model Intercomparison Project, Phase 5 (CMIP5; Taylor et al. (2012)), and at the same time the solar constant is abruptly reduced by an amount tuned to maintain top of atmosphere (TOA) energy balance and therefore keep the global mean temperature approximately at preindustrial levels. Besides providing an important theoretical underpinning to the consideration
of solar geoengineering scenarios, the G1 experiment is helpful for improving our fundamental understanding of how the climate responds differently to solar forcings, which operate in the shortwave (SW) part of the radiative spectrum, versus greenhouse gas forcings, which operate in the longwave (LW), and how linear the response is to combinations of SW and LW forcings. This can help us understand paleoclimates in which the sun was weaker (Feulner, 2012), attribution of climate change to anthropogenic as opposed to solar forcings (Santer et al., 2003), and the response of the climate to non-solar SW forcings
such as aerosol forcings (Salzmann, 2016).

An interesting question related to G1 is the amount of solar constant reduction $|\Delta S_0|$ required to compensate for the $CO_2$ increase, which has implications for the scale of the solar geoengineering intervention that would be required. This varies between about 3%-5% depending on the model (Table 1). (For convenience, we hereafter drop the absolute value symbol and use $\Delta S_0$ to refer to the solar constant reduction, keeping in mind that the sign of the change is always negative in this context.)
One might intuitively expect that $\Delta S_0$ would be greater in models with greater equilibrium climate sensitivity (ECS), which is the amount of global mean temperature change that occurs after $CO_2$ is doubled and the climate adjusts to restore top of atmosphere energy balance. However, a scatter plot of $\Delta S_0$ versus ECS for each model (Figure 1a) shows that there is actually no correlation (correlation coefficient $r = 0.02$) between these quantities. This makes sense if climate sensitivity is mainly determined by feedbacks on global mean temperature change, as has been found in CMIP5 models (Vial et al., 2013), since
the feedbacks will, at least to a first order, work just as well to reverse a warming effect when an equal and opposite radiative forcing is applied.





**Table 1.** Models included in this study, with references, institutions, solar constant reduction in the G1 experiment ($\Delta S_0$), and global mean surface air temperature change in G1 - piControl ($\Delta T$). All have a full dynamical ocean coupled to the atmosphere.

| Model | Reference | Institution | $\Delta S_0$ | $\Delta T$ (K) |
|---|---|---|---|---|
| BNU-ESM | Ji et al. (2014) | Beijing Normal University | 4.4% | 0.025 |
| CanESM-2 | Arora et al. (2011) | Canadian Centre for Climate Modeling and Analysis | 4.0% | -0.013 |
| CCSM4 | Gent et al. (2011) | National Center for Atmospheric Research | 4.1% | 0.233 |
| CESM-CAM5.1-FV | Hurrell et al. (2013) | National Center for Atmospheric Research | 4.7% | -0.157 |
| CSIRO-Mk3L-1-2 | Phipps et al. (2011) | Commonwealth Scientific and Industrial Research Organization/ | 3.2% | 0.034 |
| | | Bureau of Meteorology | | |
| GISS-E2-R | Schmidt et al. (2014) | NASA Goddard Institute for Space Studies | 4.5% | -0.292 |
| HadGEM2-ES | Collins et al. (2011) | Met Office Hadley Centre | 3.9% | 0.241 |
| IPSL-CM5A-LR | Dufresne et al. (2013) | Institut Pierre Simon Laplace | 3.5% | 0.109 |
| MIROC-ESM | Watanabe et al. (2011) | Atmosphere and Ocean Research Institute (The University of Tokyo), | 5.0% | -0.065 |
| | | National Institute for Environmental Studies, | | |
| | | and Japan Agency for Marine-Earth Science and Technology | | |
| MPI-ESM-LR | Giorgetta et al. (2013) | Max Planck Institute for Meteorology | 4.7% | -0.011 |
| NorESM1 | Bentsen et al. (2013) | Bjerknes Centre for Climate Research, | 4.0% | -0.044 |
| | | Norwegian Meteorological Institute | | |

For BNU-ESM, we are using a new realization, r3i1p1, that has a greater solar constant reduction and better compensates global mean temperature than the original. Two models that originally participated in G1, EC-Earth and HadCM3, are excluded from our analysis because many of the output fields necessary for this study were not available.

A more accurate prediction of the value of $\Delta S_0$ in G1 is provided by a simple formula based on matching the reduction in outgoing LW radiation (OLR) from the $CO_2$ increase with a reduction in the absorbed SW radiation:

$$\Delta S_0 = 4 \times \frac{F_{4\mathrm{xCO2,eff}}}{1 - \alpha} \tag{1}$$

or, in percentage terms,

5 $$\Delta S_0(\%) = \left( 4 \times \frac{F_{4\mathrm{xCO2,eff}}}{1 - \alpha} \right) \times \frac{100\%}{1361 \ \mathrm{W \ m^{-2}}} \tag{2}$$

where $F_{4\mathrm{xCO2,eff}}$ is the effective radiative forcing from a $CO_2$ quadrupling and $\alpha$ is the planetary albedo, with $F_{4\mathrm{xCO2,eff}}$ and $\alpha$ both being model-dependent. This equation was used by the modeling groups to create an initial guess for $\Delta S_0$, later tuned using successive 10-year GCM runs to obtain net TOA radiation of less than 0.1 W m$^{-2}$. This achieved a global mean temperature within 0.3 K of that in the CMIP5 preindustrial control (piControl) experiment in all the models included in our

10 study (Table 1). The technical specifications for G1 (Kravitz et al., 2011a) actually say to use the "steady state net radiation (TOA) difference" between abrupt4xCO2 and piControl as the radiative forcing in Equation 1, but this does not make sense because at steady state net TOA radiation is zero. Schmidt et al. (2012), describing the process for G1, state that effective radiative forcing, calculated by regressing net TOA radiative flux against global mean temperature change in abrupt4xCO2 and



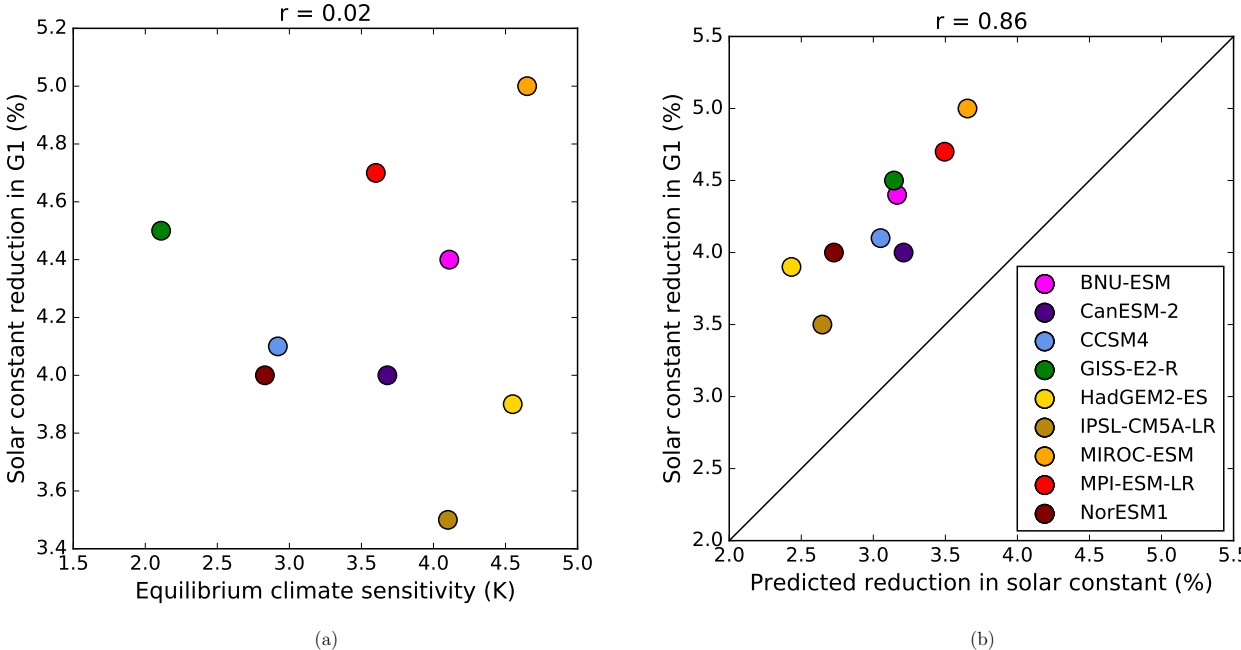

**Figure 1.** Percent solar constant reduction for models running the G1 experiment, versus (a) equilibrium climate sensitivity in the models, from Sherwood et al. (2014), and (b) solar constant reduction predicted by Equation 2, based on effective radiative forcing values from Sherwood et al. (2014) and planetary albedo values from Kravitz et al. (2013a). CESM-CAM5.1-FV and CSIRO-Mk3L-1-2 are excluded from this figure because these models were not included in Sherwood et al. (2014).

taking the intercept (Gregory et al., 2004; Gregory and Webb, 2008), was used for the initial guess in Equation 1. The effective radiative forcing incorporates rapid adjustments of the atmosphere's temperature and humidity profiles, cloud properties, and surface albedo to the $CO_2$ increase, but does not include temperature-dependent feedbacks. Strictly speaking, Equation 1 calculates the solar constant reduction that would balance the instantaneous $CO_2$ increase if atmospheric properties were 5 allowed to adjust to the $CO_2$ increase but not to the solar constant reduction. Therefore, using it to predict $\Delta S_0$ amounts to assuming that the solar forcing does not cause its own rapid adjustments.

Figure 1b compares the solar constant reduction predicted by Equation 1 to the reduction that actually achieved TOA energy balance under increased $CO_2$ after tuning. The correlation, $r = 0.86$, is much better than for ECS, and indicates that $CO_2$ forcing and planetary albedo determine $\Delta S_0$ to a first order. However, for every model, the actual $\Delta S_0$ is greater than the 10 predicted value using Equation 1, as has been noted by Schmidt et al. (2012) for a subset of four models. This indicates that atmospheric and surface adjustments in response to the combined $CO_2$ and solar instantaneous forcings have a greater net warming effect on the climate than such adjustments to the $CO_2$ forcing alone, requiring a greater reduction in the solar constant to restore the global mean temperature to preindustrial.





To resolve this discrepancy, we propose using the *instantaneous* radiative forcing from the $CO_2$ increase, which is the change in OLR from the $CO_2$ increase when all atmospheric and surface properties are held constant, and explicitly accounting for the rapid adjustments to the *combined* $CO_2$ forcing and solar reduction, as expressed in the following equation:

$$\Delta S_0 = 4 \times \frac{F_{\text{4xCO2,inst}} + \sum \Delta R_X}{1 - \alpha} \qquad (3)$$

where $F_{\text{4xCO2,inst}}$ is the instantaneous radiative forcing from the $CO_2$ quadrupling and $\Delta R_X$ represents the change in net downward TOA radiation associated with adjustments of various atmospheric and surface physical properties $X$ to the combined forcings, following the notation of Zhang and Huang (2014). The various $\Delta R_X$ can be calculated using existing tools developed for determining radiative adjustments and feedbacks. Since there is no global mean temperature change in G1 by design (and approximately none in practice), it is appropriate to continue to refer to the changes in TOA radiative balance resulting from changes in various physical properties of the atmosphere and surface as "adjustments" and not "feedbacks" even though the time scale is no longer rapid after 50 years. Kravitz et al. (2013b) plotted time series of changes in the components of the surface energy budget in G1 (their Figure 1) and found that the fluxes change little after the first year, so the time scale of the adjustment is not important for our purposes.

This study examines changes in atmospheric temperature, specific humidity, cloud fraction, and surface albedo in G1, and quantifies the radiative effects of these changes in order to test the hypothesis that the solar constant reduction can be predicted using Equation 3 and understand why the required $\Delta S_0$ is greater than that predicted using effective $CO_2$ forcing. We also explore the physical reasons for the changes in atmospheric properties, particularly cloud properties, which have been found to strongly affect meridional energy transport changes in G1, with implications for regional temperature and precipitation responses (Russotto and Ackerman, 2018). The changes in atmospheric properties, including clouds, are plotted and discussed in Section 2. Section 3 quantifies the radiative effects of these surface ant atmospheric adjustments to the G1 forcing. Section 4 examines the global means of these adjustments to see which are most important and how well Equation 3 predicts the required solar forcing in G1. In Section 5 we summarize our results and discuss implications for future research on geoengineering and solar climate forcings.

## 2   Changes in the physical state of the atmosphere

To understand the physical basis for the radiative adjustments calculated in later sections, in this section we show changes in atmospheric temperature, specific humidity, and cloud fraction that occur in the G1 experiment relative to preindustrial conditions. Throughout the paper we show averages over 40 year time periods: years 11-50 of the G1 simulation, to avoid incorporating transient effects that occur in the first ten years into averages, and years 1-40 of the piControl simulation, except where otherwise noted.

Figure 2 shows the zonal mean temperature change for G1 minus piControl in each of the 11 models listed in Table 1. Several features common to all models are apparent. First, while the global mean surface air temperatures are all within 0.3 K of preindustrial (Table 1), all of the models exhibit surface cooling in the tropics and warming in the polar regions. This





**Figure 2.** Zonal mean temperature change for G1 minus piControl in each model as a function of pressure.

phenomenon has long been noted as a feature of climate model experiments with the G1 setup (e.g. Govindasamy et al., 2003; Kravitz et al., 2013a), and is due to the imposition of a net negative forcing in the tropics and a net positive forcing at the poles (Russotto and Ackerman, 2018). However, cooling dominates when considering the atmosphere as a whole. The tropical mid-to-upper troposphere cools more than the surface does, because the tropical temperature profile tends to follow a moist

5   adiabat (e.g. Wetherald and Manabe, 1975), so that slight cooling at the surface leads to greater cooling aloft. The cooling of the tropical upper troposphere mirrors the effect that happens in global warming, where the upper troposphere warms more than the surface and emits more LW radiation, leading to a negative climate feedback known as the lapse rate feedback. In the case of G1, reduced LW emission from the atmospheric cooling has a warming effect on the planet; we quantify this effect using radiative kernels in Section 3.2.

10   Most models have an area of reduced cooling or even warming in the tropics near 100 hPa. This corresponds to the location of the tropical tropopause layer (TTL), an area in the tropics between about 70 and 150 hPa with properties of both the



**Figure 3.** Zonal mean change in the natural log of specific humidity for G1 minus piControl in each model as a function of pressure.

troposphere and stratosphere (Fueglistaler et al., 2009). The detailed vertical structure of temperature changes here may have to do with complex interactions between local temperature, humidity, and cloud properties. Another notable feature of the temperature change is the cooling of the stratosphere. An increase in carbon dioxide concentration cools the stratosphere, due to increased emission of LW radiation to space (Manabe and Wetherald, 1975), and a decrease in the solar constant also cools

5 the stratosphere because it reduces the amount of ultraviolet radiation absorbed by ozone. The stratospheric cooling effect from the solar constant reduction is about an order of magnitude smaller than that from the $CO_2$ quadrupling (Govindasamy et al., 2003).

Figure 3 shows the change in the log of specific humidity between G1 and piControl in each model. We use a log scale because it makes it easier to visualize changes in specific humidity that occur over multiple orders of magnitude, and because

10 log humidity changes are used in the water vapor radiative kernel calculations described in Section 3.2. Most of the troposphere becomes drier in G1 in all models, consistent with the large-scale cooling absent significant changes in relative humidity. Since



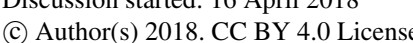

**Figure 4.** Zonal mean change in cloud fraction for G1 minus piControl in each model as a function of pressure, or height for HadGEM2-ES. To help comparisons with other models, the vertical axis for HadGEM2-ES is scaled according to $e^{-z/8000 \text{ m}}$ (where $z$ is height), which is approximately proportional to pressure.

water vapor is a strong greenhouse gas, this drying has a cooling effect on the planet, which we quantify in Section 3.2. Most models show moistening in the polar regions at low altitudes, consistent with the warming there, although the moistening is typically confined to smaller areas than the warming, indicative of a slight decrease in relative humidity at the poles (cf. Figure 5 of Smyth et al. (2017)). Interestingly, stratospheric water vapor decreases in most models, but it increases in the three models,

5   BNU-ESM, CSIRO-Mk3L-1-2, and IPSL-CM5A-LR, that have warming in the TTL (albeit this moistening is mostly confined to the northern hemisphere in the IPSL model). This is consistent with stratospheric humidity being set by temperatures in the TTL, through which air travels to reach the stratosphere as part of the Brewer-Dobson Circulation (e.g. Brewer, 1949; Newell and Gould-Stewart, 1981).



Figure 4 shows the zonal mean changes in cloud fraction in each of the models for G1 - piControl. Unlike atmospheric temperature and humidity, cloud fraction model output in CMIP5 and GeoMIP was archived on the native model vertical grid instead of a set of standardized pressure levels. Most of the GeoMIP models use hybrid sigma pressure coordinates, with the exceptions of GISS-E2-R, which uses pressure coordinates, and HadGEM2-ES, which uses hybrid sigma height

coordinates. To enable direct comparisons with the temperature and humidity changes and radiative kernel calculations, we have regridded the cloud fraction output to the standard CMIP5 pressure levels, or to a fixed height grid for HadGEM2-ES. Conversion from hybrid sigma to pressure or height coordinates was done using a Python function (see "Code and data availability" below) based on the algorithm used in the "convert_sigma_to_pres" Matlab function by Vimont, available at http://www.aos.wisc.edu/~dvimont/matlab/. Since surface pressure output (required for the hybrid sigma pressure regridding)

was only available for the last 50 years of the piControl simulation for CSIRO-Mk3L-1-2, we have used the last 40 years of this simulation as the control case for cloud fraction for this model, instead of the first 40 years.

In their study of four models running G1, Schmidt et al. (2012) noted that all four had a reduction in low cloud fraction, while high clouds had an inconsistent change. Figure 4 shows that an overall reduction of low cloud fraction occurs in all 11 models included in this study. For high clouds, we also find an inconsistent response, but overall high cloud fraction increases in most

models. Some models, especially those in which the TTL warms (Figure 2), have a decrease in high cloud fraction in the TTL, and two of them, CSIRO-Mk3L-1-2 and IPSL-CM5A-LR, have an overall decrease in high cloud fraction. Since low clouds primarily have a cooling effect on the climate due to their strong SW reflection, a reduction in low clouds would result in a warming effect that would partially counteract the cooling from solar geoengineering. An increase in high cloud fraction would also be expected to have a warming effect on the planet by reducing LW emission to space, although other variables, such as

cloud height, are more important to the LW effect of cloud changes in global warming simulations (Zelinka et al., 2012b). High clouds also interact with local temperature and humidity profiles in complex ways; for example, the reduced high cloud fraction in the IPSL model appears to warm the atmosphere immediately above, due to a reduction in LW emission from the cloud tops. We quantify the TOA SW and LW effects of the changes in cloud properties in Sections 3.1 and 3.2, respectively. In many models there is an increase in clouds in the stratosphere over Antarctica, likely due to the stratospheric cooling. Two

models, HadGEM2-ES and MIROC-ESM, have a dipole in cloud fraction changes in the upper troposphere, corresponding to northward and southward shifts, respectively, of the intertropical convergence zone (ITCZ) in these models (Smyth et al., 2017; Russotto and Ackerman, 2018).

To get a sense of the zonally asymmetric spatial patterns of cloud fraction changes and to better understand areas of inter-model consensus and disagreement, we plot in Figure 5 maps of the multi-model mean changes in low, middle, and high cloud

fraction for G1 - piControl. Within the ranges for low, middle, and high clouds, we assume random overlap between adjacent layers of the common pressure grid. We use 680 hPa as the boundary between low and middle clouds and 440 hPa as the boundary between middle and high clouds, following the standards for the International Satellite Cloud Climatology Project (ISCCP; see Figure 2 of Rossow and Schiffer (1999)), or 3250 m and 6500 m in the case of HadGEM2-ES, which roughly correspond to these pressure levels in the 1976 Standard Atmosphere (NOAA, 1976). These plots, and all subsequent multi-

model mean maps, show stippling where fewer than all but 2 of the included models agree on the sign of the change, so that





**Table 2.** Global mean changes in low, middle and high cloud fraction in G1 minus piControl.

| Model | Cloud fraction change (%) | | |
|---|---|---|---|
| | low | middle | high |
| BNU-ESM | -0.60 | 0.26 | 0.91 |
| CanESM-2 | -1.59 | -0.14 | 0.69 |
| CCSM4 | -1.54 | -0.15 | 1.19 |
| CESM-CAM5.1-FV | -1.51 | 0.03 | 1.38 |
| CSIRO-Mk3L-1-2 | -0.71 | -0.51 | -0.57 |
| GISS-E2-R | -1.04 | 0.13 | 1.34 |
| HadGEM2-ES | -1.38 | -0.19 | 0.44 |
| IPSL-CM5A-LR | -0.74 | 0.03 | -1.12 |
| MIROC-ESM | -1.60 | -0.02 | 0.76 |
| MPI-ESM-LR | -1.03 | -0.05 | 1.31 |
| NorESM1 | -1.63 | -0.22 | 1.12 |

unstippled areas indicate robust changes. Since this agreement could happen by chance in isolated areas, we focus on areas with apparent spatial structure or a physical reason why we might expect a change. For all multi-model mean maps, corresponding maps for each of the individual models are available in the Supplemental Information. Global mean cloud fraction changes for the individual models are shown in Table 2.

The reduction in low cloud fraction (Figure 5a) is widespread, occurring over most ocean areas except for regions close to the equator and poles, and over most non-desert land areas. Middle clouds (Figure 5b) have fewer areas with robust changes, but there is a reduction in the cloud fraction on either side of the equator over the Atlantic and Pacific and over the equatorial Indian ocean. This may be related to a narrowing of the annual mean tropical precipitation maximum (see Fig. 5 of Tilmes et al., 2013), which may be due in part to a reduction in the seasonal migration of the ITCZ (Smyth et al., 2017). For high

clouds (Figure 5c), there are few areas with robust changes, but there is a notable increase in high clouds over the equator, in some subtropical regions (around 30° N and S), and over the poles, particularly Antarctica. Figure 4 shows that the high cloud increases over the poles are mostly in the stratosphere.

   Without additional experiments varying potential drivers of cloud changes, it is difficult to prove definitively the causes for the changes in cloud fraction. However, it is possible to gain some insight into the reasons for changes in low cloud

fraction over the ocean by plotting several variables that are correlated with low cloud fraction in observations. These include lower-tropospheric stability (LTS), defined as the difference in potential temperature between 700 hPa and the surface (Klein and Hartmann, 1993), and estimated inversion strength (EIS), a metric of the temperature inversion at the top of the marine boundary layer. EIS is defined as (Wood and Bretherton, 2006, Eq. 4):

$$\text{EIS} = \text{LTS} - \Gamma_m^{850}(z_{700} - \text{LCL}) \tag{4}$$




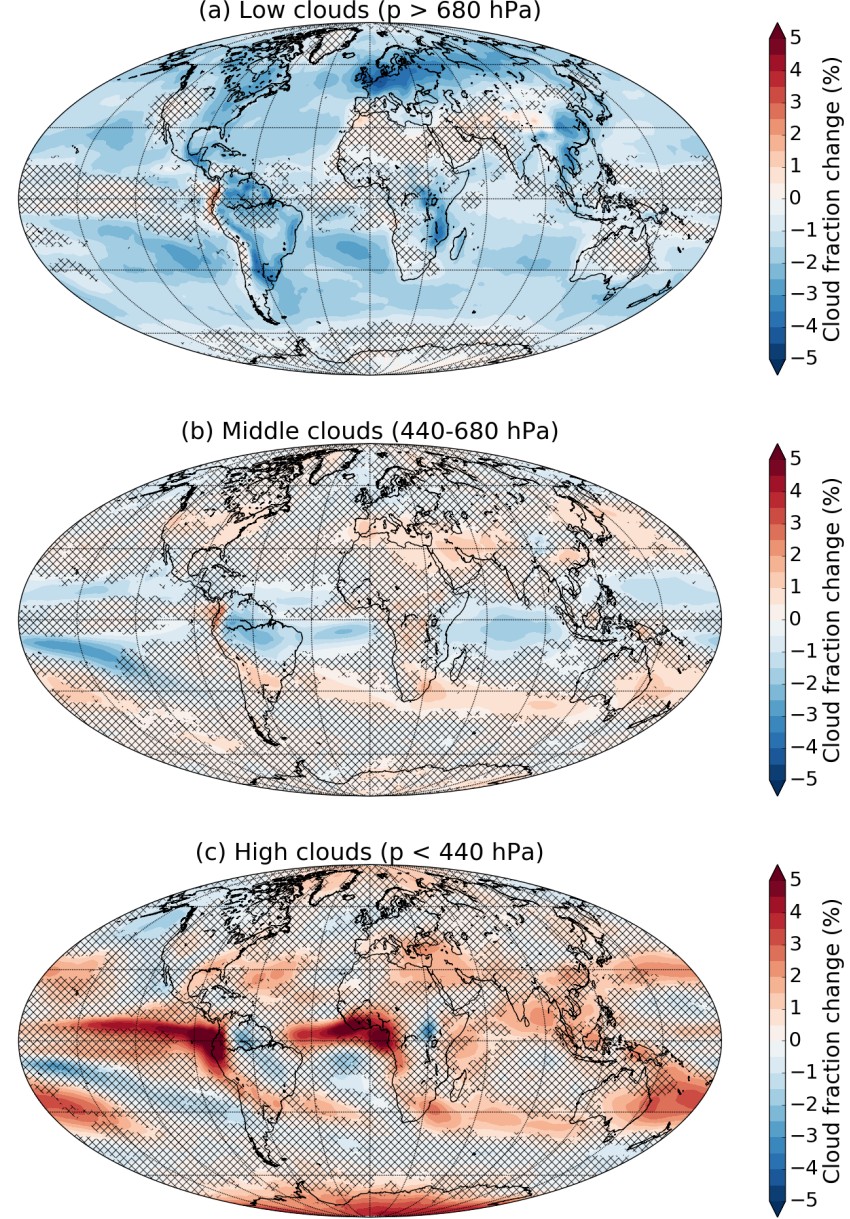

**Figure 5.** Multi-model mean changes in low (a), middle (b) and high (c) cloud fraction for G1 - piControl. Hatching indicates areas where fewer than 9 of the 11 models agree on the sign of the change.

where $\Gamma_m^{850}$ is the moist adiabatic lapse rate at 850 hPa, $z_{700}$ is the height of the 700 hPa surface, and LCL is the lifting condensation level.



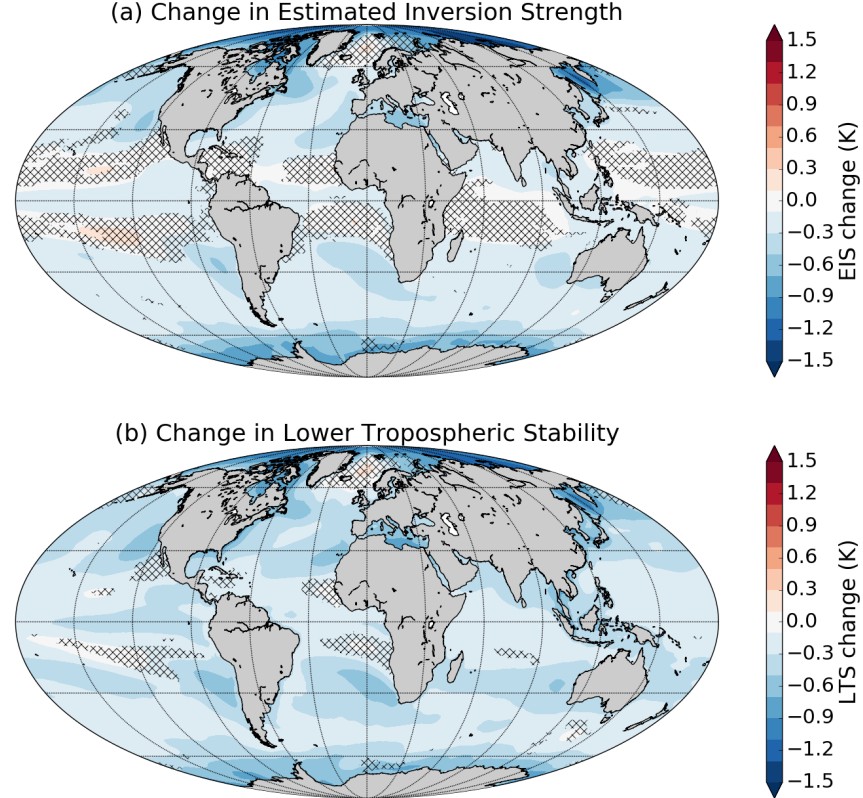

**Figure 6.** Multi-model mean changes in EIS (a) and LTS (b) for G1 - piControl. Hatching indicates areas where fewer than 7 of 9 models agree on the sign of the change. CSIRO-Mk3L-1-2 and MPI-ESM-LR models are excluded from this plot because near-surface specific humidity output, which is required to calculate EIS, was not available.

Figure 6 shows the changes in EIS (a) and LTS (b) for G1 - piControl. Both of these quantities generally decrease across most of the ocean, except for some regions centered near 15° N and S. The reduction in EIS is generally smaller than the reduction in LTS (due to the correction for the moist adiabatic temperature profile), but is still widespread. A reduction of the strength of the inversion at the top of the boundary layer would be expected to reduce low cloud fraction by encouraging

5    mixing of dry air into the boundary layer, so the reduction in EIS over the ocean is a likely explanation for the reduction in low cloud fraction there. Stability metrics are included in low cloud fraction schemes in many models, and those that use the Slingo (1987) scheme, such as CCSM4 and NorESM1-M, have an explicit dependence of low cloud fraction on stability. However, the robustness of the reduction in low cloud fraction in G1 indicates that it is not the result of the idiosyncrasies of any one cloud fraction scheme.

10    Besides changes in stability metrics, other factors that have been suggested as explaining changes in marine stratocumulus cloud fraction under global warming conditions in large-eddy simulation models include reduced LW radiative cooling from



cloud tops due to increased $CO_2$ and $H_2O$ concentrations; decreased subsidence above the boundary layer; and increased sea surface temperatures (Bretherton, 2015). Qu et al. (2014) analyzed changes in marine stratiform cloud fraction in CMIP3 and CMIP5 global warming experiments, and found a reduced low cloud fraction in most models, which they attributed to an increase in sea surface temperature (SST). While EIS increased, which would promote increased cloud fraction, this was not

enough to compensate for the SST increase in the global warming scenarios. In G1, SST changes little (and in fact decreases slightly in the tropics and subtropics (Hong et al., 2017, Fig. 1)), leaving EIS to dominate changes in low cloud fraction over the ocean. It does not appear that cloud top radiation or subsidence could be responsible for the widespread low cloud reduction. The mechanism of reduced radiative cooling from cloud tops would be much weaker for G1 than for global warming if at all present because, while $CO_2$ concentrations have increased, water vapor concentrations have decreased. We have not tried to

quantify how these fluxes have changed in G1 since LW radiative fluxes at the top of the boundary layer were not included in the GeoMIP model output archive. Meridional stream function anomaly plots for G1 minus piControl (Smyth et al., 2017; Guo et al., 2018) show that while some areas have anomalous subsidence, others have anomalous rising motion, and these regions are not consistent between models.

Qu et al. (2014) attribute the increase in EIS in global warming experiments to greater surface warming over the continents

and the tropical western Pacific warm pool relative to the rest of the ocean; the warmed air is then advected over the tops of the marine stratocumulus fields. However, a reverse version of this mechanism does not seem to be at work in G1 because cooling is more robust over the ocean than over land (Kravitz et al., 2013a, Figure 2). It is also important to keep in mind that there are different metrics of stability that are useful for different parts of the atmosphere and for different types of clouds. Kravitz et al. (2013b) argued that any cloud cover changes in G1 would be due in part to increases in atmospheric stability, but in our study

it appears to be a decrease in stability that is most relevant to the low cloud reduction. Another metric of stability, the rate of increase of equivalent potential temperature with height, does in fact increase in G1 relative to piControl, as shown in Figure 8 of Kravitz et al. (2013a). So, even as the atmosphere has gotten less stable in G1 with respect to boundary layer turbulence, it has gotten more stable with respect to deep convection. To better understand the reasons for the changes in clouds, it would be useful to further investigate the effects of $CO_2$ and solar forcings on potential and equivalent potential temperature profiles.

Over land, existing research suggests that the reduction in low cloud fraction in G1 is a result of the physiological responses of plants to increased $CO_2$, as represented in the models' dynamic vegetation schemes. Cao et al. (2010) ran GCM simulations in which the $CO_2$ concentrations experienced by plants were doubled while the radiative fluxes were held constant, and found that low cloud fraction decreased in many vegetated land areas (see their Figure 1, central panel). The low cloud fraction decrease in the Cao et al. study is strongest in South America, eastern North America, southeast Asia, southeast Africa, and

western Europe, which are the same areas of reduced low cloud cover in G1. The mechanism is that, when $CO_2$ concentrations are higher, plants' stomata do not need to open as much to take in the same amount of $CO_2$, leading to less transpiration of water from the plants (Field et al., 1995). This causes a reduction in near-surface relative humidity over land, seen in both Cao et al. (2010, Figure 2) and G1 (Smyth et al., 2017, Figure 5), which reduces the cloud fraction. In addition to plant physiology, it is possible that some of the reduction in relative humidity and cloud fraction over land in G1 is due to a reduction in evaporation

directly caused by the reduction in surface SW radiation. The balance between these two quantities explains the reduction in





global mean precipitation in G1 (Kravitz et al., 2013b), since precipitation must balance evaporation, suggesting that a similar mechanism may affect cloud fraction. Over the ocean, however, near-surface relative humidity increases in G1 in most areas, despite the reduction in evaporation (Smyth et al., 2017), implying that evaporation changes are not the reason for the low cloud changes there.

## 3   Radiative effects

### 3.1   SW radiative effects

To calculate the SW radiative effects of changes in clouds and other atmospheric and surface properties, we use the approximate partial radiation perturbation (APRP) method introduced by Taylor et al. (2007), which is based on a single-layer radiative transfer model of the atmosphere that can be expressed analytically and requires as inputs only the monthly mean surface and
TOA radiative fluxes and total column cloud fraction outputs from the GCMs. APRP shows the radiative effects of physical changes in clouds, accounting for cloud masking effects, in which the differences between clear-sky and all-sky fluxes change in response to forcing without changes in the clouds themselves. The calculations shown here have previously been used as inputs to energy balance model simulations to understand the effects of changes in clouds and surface albedo on atmospheric energy transport in G1 (Russotto and Ackerman, 2018).

Figure 7 shows the multi-model mean change in net downward SW radiative flux at the TOA due to changes in clouds (Figure 7a), non-cloud atmospheric scattering and absorption (Figure 7b), and surface albedo (Figure 7c), calculated using APRP. Global mean radiative adjustments for the individual models in the SW and LW are shown in Table 3, which will be referred to in the discussion of the required solar forcing in G1 in Section 4. Clouds generally have a robust and widespread warming effect in the SW, in locations that closely correspond to the areas of reduced low cloud fraction shown in Figure 5a.
The non-cloud atmosphere effects are very weak by comparison in the multi-model mean, but there are several models with appreciable positive values for this adjustment. Maps of this adjustment for the individual models (Figure S7) show that for HadGEM2-ES, it appears to be related to a reduction in atmospheric dust, since most of the warming effect occurs over and downwind of deserts; in IPSL-CM5A-LR, the effect is relatively spatially uniform but slightly stronger in higher latitudes. For surface albedo, there are warming effects in high latitudes from decreases in sea ice and snow cover associated with the residual
polar warming in G1. There are also some warming effects in lower latitudes near desert regions, such as in the Sahel region; this may have to do with vegetation effects. There are several small regions, such as Tibet, with increases in surface albedo, presumably due to increased snow cover as a result of surface cooling there (*cf.* Figure 2 of Kravitz et al. (2013a)). Surface albedo effects are strong in some locations, such as the Sea of Okhotsk, but the relatively small area over which surface albedo changes can occur limits their importance in the global mean.





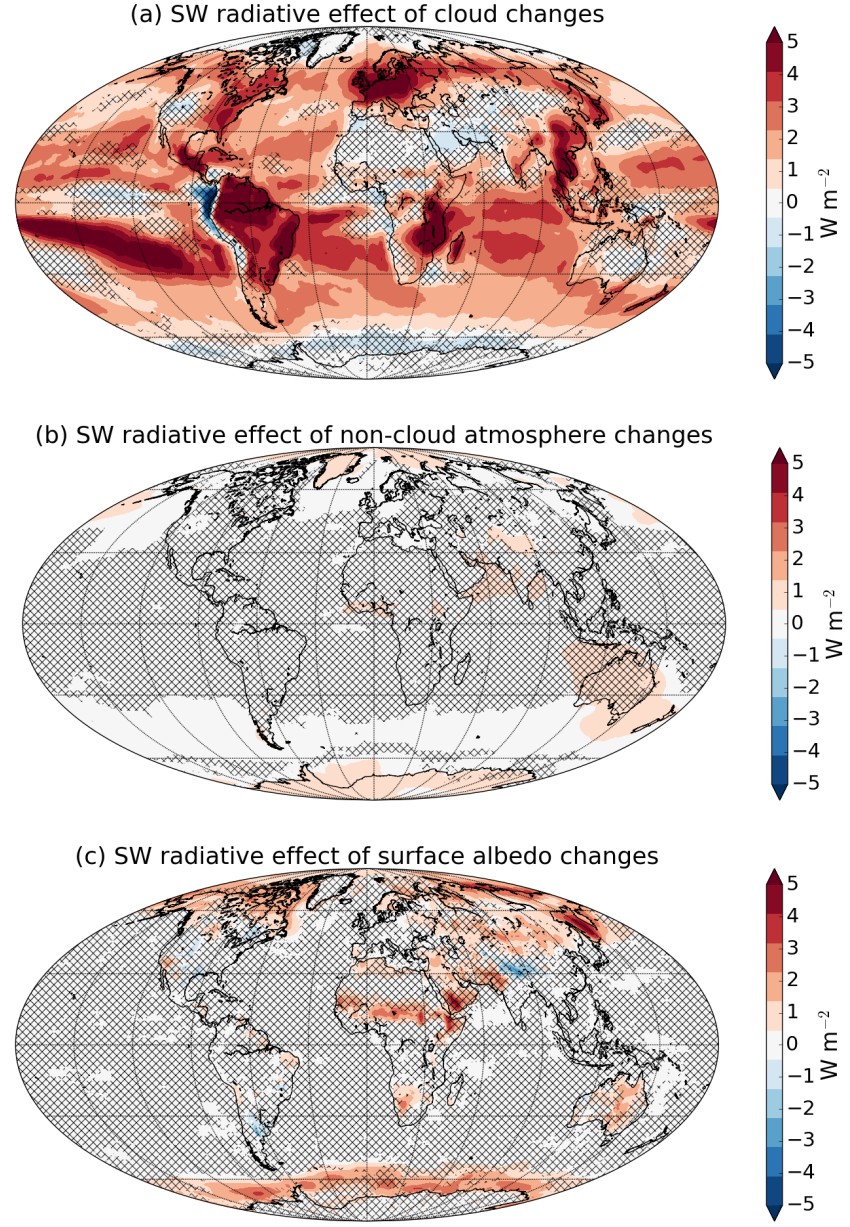

**Figure 7.** Multi-model mean change in net downward SW radiation at the TOA in G1 - piControl due to changes in cloud properties (a), non-cloud atmospheric absorption and scattering (b), and surface albedo (c), calculated using APRP method (Taylor et al., 2007). Hatching indicates areas where fewer than 7 of 9 models agree on the sign of the change. CSIRO-Mk3L-1-2 and GISS-E2-R models are excluded because not all fields necessary for APRP were correctly archived.





**Table 3.** Global mean radiative adjustments in G1 minus piControl, and excess and total solar forcing in G1, in W m$^{-2}$. Positive values indicate a warming effect (increase in absorbed SW radiation or decrease in OLR), except for solar forcing where positive values represent a cooling. SW adjustments correspond to multi-model means plotted in Figure 7. LW adjustments correspond to multi-model means plotted in Figures 8 and 9, with sign flipped for Figure 8. "Sum" is the sum of all the SW and LW adjustments. $F_{\text{excess}}$ is calculated using Equation 6 and represents the actual instantaneous solar forcing ($F_{\text{solar}}$) in G1 minus that predicted based on $CO_2$ effective or instantaneous forcing. $F_{\text{solar}}$ represents the total instantaneous solar forcing calculated from theory (Equation 8) or actually used in G1 (Equation 9).

| Model | SW adjustments | | | LW adjustments | | | | Sum | $F_{\text{excess}}$ | | $F_{\text{solar}}$ | |
|---|---|---|---|---|---|---|---|---|---|---|---|---|
| | cloud | non-cloud | surface | $T_a$ | $T_s$ | $H_2O$ | cloud | | Eff. | Inst. | theory | actual |
| BNU-ESM | 1.36 | 0.05 | 0.51 | 2.94 | 0.08 | -0.78 | -0.08 | 4.08 | 2.95 | — | — | 10.51 |
| CanESM-2 | 1.44 | 0.41 | -0.04 | 3.03 | 0.07 | -1.04 | -0.26 | 3.60 | 1.90 | 4.00 | 9.20 | 9.60 |
| CCSM4 | 2.09 | -0.05 | 0.28 | 2.53 | -0.08 | -0.84 | 0.13 | 4.05 | 2.55 | 4.44 | 9.56 | 9.95 |
| CESM-CAM5.1-FV | 0.71 | -0.09 | 0.87 | 3.94 | 0.18 | -1.39 | 0.30 | 4.52 | — | — | — | 11.26 |
| CSIRO-Mk3L-1-2 | — | — | — | 2.16 | 0.03 | -0.52 | -0.24 | — | — | — | — | — |
| GISS-E2-R | — | — | — | 4.88 | 0.21 | -1.78 | -0.07 | — | — | — | — | 10.79 |
| HadGEM2-ES | 1.05 | 1.07 | 0.50 | 2.66 | -0.05 | -0.87 | -0.15 | 4.21 | 3.56 | 3.91 | 9.76 | 9.46 |
| IPSL-CM5A-LR | 1.32 | 1.21 | 0.15 | 2.08 | -0.05 | -0.52 | -0.86 | 3.35 | 2.01 | 3.85 | 7.75 | 8.25 |
| MIROC-ESM | 3.29 | 0.06 | 0.02 | 3.44 | 0.10 | -1.11 | -0.58 | 5.22 | 3.15 | — | — | 11.69 |
| MPI-ESM-LR | 2.63 | -0.00 | 0.17 | 3.41 | 0.07 | -1.10 | -0.54 | 4.63 | 2.86 | — | — | 11.16 |
| NorESM1 | 2.07 | -0.20 | 0.05 | 2.88 | 0.08 | -0.97 | -0.10 | 3.82 | 3.00 | 3.87 | 9.34 | 9.42 |

## 3.2 LW radiative effects

The technique of radiative kernels (Held and Soden, 2006; Soden et al., 2008; Shell et al., 2008) was developed to quantify LW radiative adjustments and feedbacks using standard monthly mean climate model output. These kernels consist of matrices of the partial derivatives of OLR with respect to changes in surface temperature, atmospheric temperature, specific humidity, and greenhouse gas concentration as a function of latitude, longitude, month and (where applicable) pressure, calculated using offline calculations with a particular GCM's radiative transfer code. Radiative kernels have been developed based on a variety of GCMs, including GFDL AM2 (Soden et al., 2008), CAM3 (Shell et al., 2008), MPI-ESM-LR (Block and Mauritsen, 2013), and CESM-CAM5 (Pendergrass et al., 2018).

We have applied the Shell et al. (2008) radiative kernels to the G1 ensemble. The choice of model used to generate the kernels has been shown to have little effect on the results (Soden et al., 2008). After regridding the kernels to the latitude and longitude grid of each GCM, we multiplied them by the changes in temperature and the log of specific humidity, normalized by the standard anomaly used to compute the kernels (1 K for the surface and atmospheric temperature kernels, and the change in log specific humidity associated with a 1 K warming at constant relative humidity for the water vapor kernel), in order to compute the change in OLR associated with the changes in each of these quantities for G1 - piControl. We summed the





OLR changes from each vertical level in order to get overall radiative adjustments from column temperature and water vapor changes, and we used the annual mean of the monthly results for our analysis.

Figure 8 shows multi-model mean changes in OLR for G1 - piControl calculated from the atmospheric temperature (a), surface temperature (b), and water vapor (c) kernels. Global means for the individual models are shown in Table 3. For the

atmospheric temperature kernel, there is a strong decrease in OLR that is widespread across the globe and robust across models. This is associated with the cooling of the atmosphere and reduced longwave emission (*cf.* Figure 2). The reduction in OLR is stronger in the tropics than in the polar regions, and is due to some combination of upper tropospheric and stratospheric cooling. We discuss the contribution of the stratospheric component in the next section. Surface temperature changes have little effect on the TOA LW radiation balance, but there is a reduction in OLR in the tropics and subtropics and an increase in the polar

regions that is consistent across models, due to the patterns of tropical cooling and polar warming at the surface. The OLR change from the surface temperature kernel is much smaller than that for atmospheric temperature because the atmosphere is not very transparent to LW radiation in most wavelengths, and because temperature changes are smaller at the surface than in the upper troposphere and stratosphere. Changes in water vapor concentration cause a robust cooling effect (increase in OLR) that partially offsets the warming effect from the atmospheric temperature kernel. The water vapor concentration decreases in

the upper troposphere (Figure 3), which increases LW emission to space by lowering the effective altitude of emission.

In addition to the quantities plotted in Figure 8, radiative kernels can also be used to calculate the effect of changes in cloud properties on OLR. This is often measured according to the change in the cloud radiative effect (CRE), which is the difference in OLR in clear-sky minus all-sky averages. However, changes in the cloud radiative effect may include cloud masking effects. We can correct the change in LW CRE for cloud masking using the difference in flux changes calculated according to clear-sky

and all-sky kernels, following Shell et al. (2008):

$$
\begin{aligned}
\Delta\mathrm{LWCRE}_{\mathrm{adjusted}} = {}& \mathrm{LWCRE}_{\mathrm{G1}} - \mathrm{LWCRE}_{\mathrm{piControl}} \\
& + (\Delta\mathrm{OLR}_{k,T} - \Delta\mathrm{OLR}_{k,T,\mathrm{clear}} + \Delta\mathrm{OLR}_{k,T_s} - \Delta\mathrm{OLR}_{k,T_s,\mathrm{clear}} \\
& + \Delta\mathrm{OLR}_{k,q} - \Delta\mathrm{OLR}_{k,q,\mathrm{clear}} + \Delta\mathrm{OLR}_{k,\mathrm{CO_2}} - \Delta\mathrm{OLR}_{k,\mathrm{CO_2},\mathrm{clear}})
\end{aligned}
\tag{5}
$$

where, in the subscripts, $k$ denotes a change in OLR calculated using a kernel, clear denotes quantities calculated using the clear-sky instead of all-sky kernels, $T$ is atmospheric temperature, $T_s$ is surface temperature, and $q$ is specific humidity. Since the Shell et al. (2008) $CO_2$ forcing kernels were for a doubling of $CO_2$, we doubled these kernels to obtain the radiative flux

changes for a $CO_2$ quadrupling.

Figure 9 shows the multi-model mean change in LW CRE calculated using Equation 5. There is a modest cooling effect in the global, multi-model mean (see also Table 3), but there are some places where there is a robust warming effect. The strongest warming effects occur near the eastern equatorial oceans, where the increase in high cloud fraction is greatest, while the strongest cooling effects occur in two belts in the eastern Pacific, which are associated with robust decreases in low and

middle cloud fraction (*cf.* Figure 5). There are also widespread cooling effects over the mid-latitude oceans, where low cloud fraction decreases. Generally, an increase in high cloud fraction would be expected to result in a warming effect, because high clouds are much cooler than the surface and are more effective at trapping LW radiation. However, in the case of G1, it





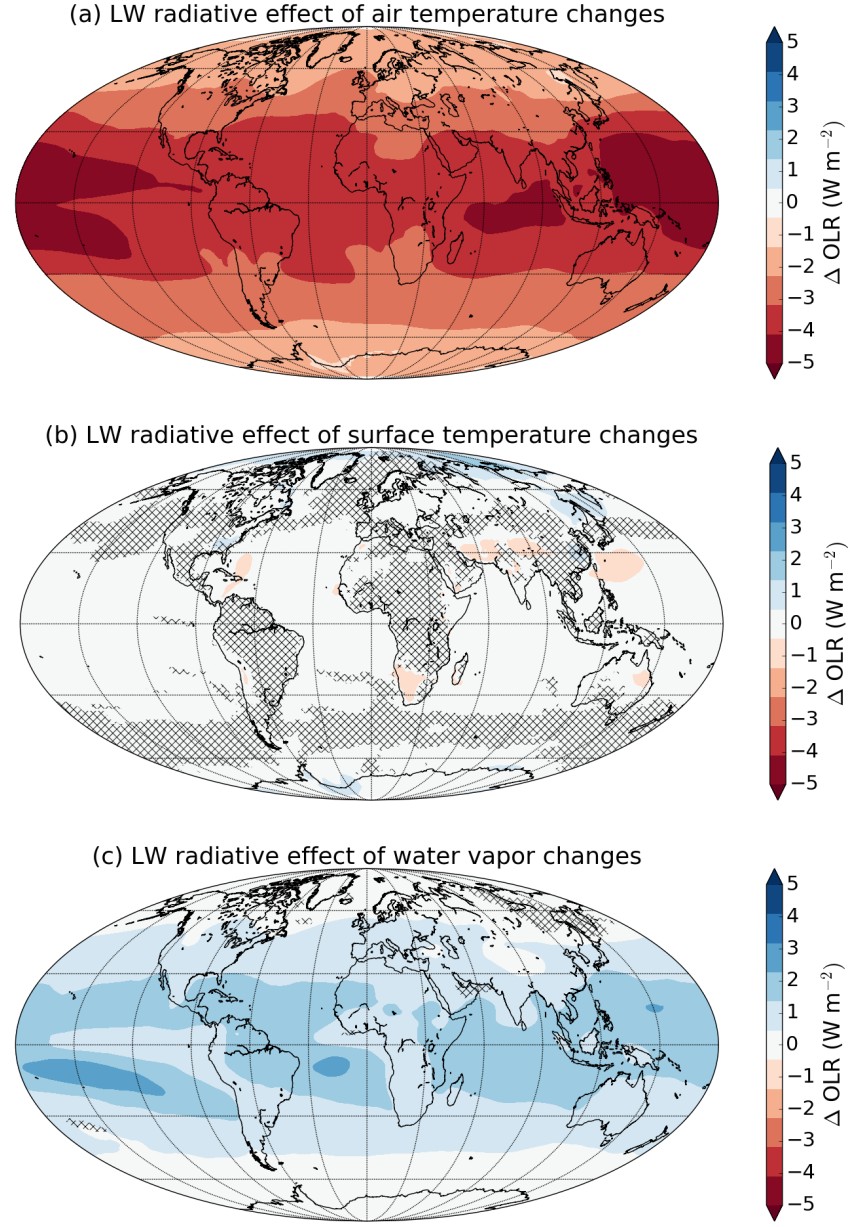

**Figure 8.** Multi-model mean change in OLR in G1 - piControl due to changes in atmospheric temperature (a), surface temperature (b), and specific humidity (c), calculated using radiative kernels (Shell et al., 2008). Hatching indicates areas where fewer than 9 of 11 models agree on the sign of the change.





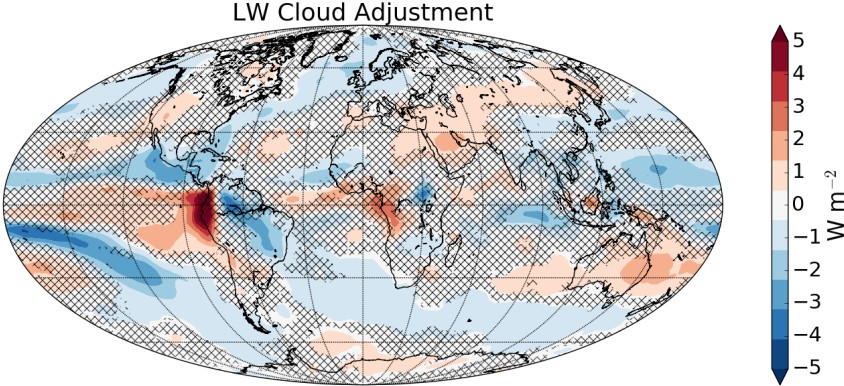

**Figure 9.** Multi-model mean change in LW cloud radiative effect in G1 - piControl, corrected for cloud masking of LW air temperature, surface temperature and water vapor adjustments and $CO_2$ forcing. Positive values indicate a decrease in OLR, *i.e.* a warming effect. Hatching indicates areas where fewer than 9 of 11 models agree on the sign of the change.

appears that the LW effect of the decrease in low cloud fraction compensates for this. Besides cloud fraction, LW radiation is also sensitive to changes in cloud height and cloud optical depth (Zelinka et al., 2012b). It may be that the global mean increase in high cloud fraction that occurs in most models has a limited effect on OLR because the new clouds being formed are optically thin; we would especially expect this in the case of polar stratospheric clouds. The radiative effects of changes in
cloud optical thickness are difficult to assess from the GeoMIP output currently available. These effects have been quantified in global warming simulations using cloud radiative kernels (Zelinka et al., 2012a), but the use of these requires cloud fraction statistics binned by optical depth and cloud top height produced by the ISCCP satellite simulator (Klein and Jakob, 1999; Webb et al., 2001) that is part of the CFMIP Observation Simulator Package (Bodas-Salcedo et al., 2011). The simulator must be run inline with each GCM, or else requires instantaneous cloud fraction output (rather than monthly means) in order to be run
retrospectively. The necessary outputs for cloud radiative kernels were saved in the Cloud Feedback Model Intercomparison Project (CFMIP; Bony et al., 2011) but not in GeoMIP. It would be useful to follow the CFMIP protocols in future GeoMIP experiments in order to allow further quantitative analysis of the changes in clouds that occur under combined SW and LW forcings.

## 4   Connections between radiative effects and required solar reduction

Having quantified the radiative effects of changes in the physical properties of the atmosphere and surface in G1, we now revisit the question of the amount of solar constant reduction required to offset the quadrupling of $CO_2$. The solar constant reduction predicted based on effective $CO_2$ radiative forcing (Equation 1) systematically underestimated the actual reduction required (Figure 1b). In this section we attempt to account for this discrepancy by comparing the amount of extra solar forcing needed with the global means of the radiative adjustments calculated in Section 3. This comparison is shown in Figure 10 for





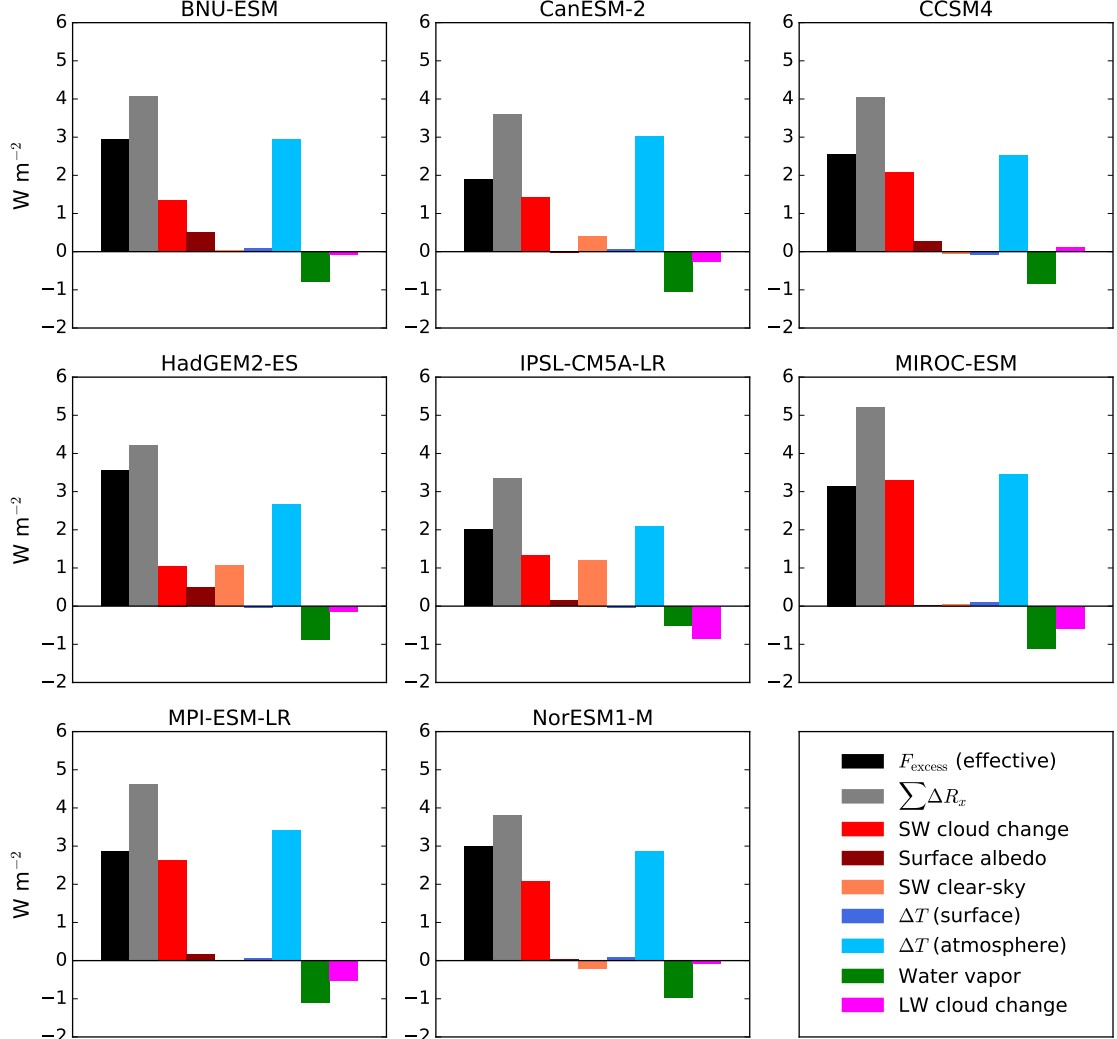

**Figure 10.** Excess required solar radiative forcing in G1 vs. that expected from effective $CO_2$ forcing (black bar), global mean SW and LW radiative adjustments (colored bars), and sum of all the radiative adjustments (gray bar), in models for which all of these quantities were calculated. For all except $F_{excess}$, positive values indicate a warming effect (increase in absorbed SW radiation or reduction in OLR). The first three colored bars correspond to the SW radiative adjustments calculated using APRP (multi-model mean maps shown in Figure 7). The three blue and green bars correspond to the LW radiative adjustments calculated using radiative kernels (multi-model mean maps shown in Figure 8). The magenta bar corresponds to the change in LW cloud radiative effect, corrected for cloud masking effects using radiative kernels (multi-model mean map shown in Figure 9).

the 8 models for which effective radiative forcing values from Sherwood et al. (2014) were available and all of the radiative





adjustments could be calculated. The excess required solar reduction, $F_{\text{excess}}$, shown in black, is calculated according to:

$$F_{\text{excess}} = (\Delta S_0(\%)_{\text{actual}} - \Delta S_0(\%)_{\text{predicted}}) \times \frac{1361 \text{ W m}^{-2}}{100\%} \times \frac{1-\alpha}{4} \qquad (6)$$

where $\Delta S_0(\%)_{\text{actual}}$ is listed in Table 1 and $\Delta S_0(\%)_{\text{predicted}}$ is calculated using Equation 2. In terms of radiative forcing, $F_{\text{excess}}$ is the difference between the actual solar forcing required in G1 and the effective forcing from the $CO_2$ quadrupling.

The relative sizes of the bars in Figure 10 are fairly similar across models. The strongest warming effect is generally from the LW atmospheric temperature adjustment, followed by the SW cloud adjustment. The only consistent cooling effect comes from the LW water vapor adjustment. Surface albedo effects are generally small, as is the SW clear-sky adjusment, with the exceptions discussed in Section 3.1. The LW surface temperature adjustment is practically negligible in all models, while the LW cloud adjustment is also small but has an inter-model range of about 1 W m$^{-2}$. The model with the greatest cooling effect

from the LW cloud adjustment, IPSL-CM5A-LR, is the model with the greatest global mean decrease in high cloud fraction, whereas most other models have an increase in high cloud fraction (Table 2).

Comparing the black and gray bars in Figure 10 shows that the sum of all the global mean radiative adjustments more than accounts for the additional solar constant reduction required to balance the $CO_2$ quadrupling, compared to the amount predicted by Equation 1. The fact that the sum of the radiative adjustments consistently overestimates $F_{\text{excess}}$ points to the fact that this

is not really a fair comparison. Rapid adjustments to a $CO_2$ quadrupling by itself, which were included in the calculation of effective $CO_2$ radiative forcing, are being double-counted, because they also show up in the radiative adjustments to the G1 combined forcing, to the extent that they are not canceled by the solar reduction.

To account for this, we plot in Figure 11 the same quantities as in Figure 10 but where the black bars are calculated using instantaneous rather than effective $CO_2$ forcing for the predicted solar constant reduction (*i.e.* using $F_{\text{4xCO2,inst}}$ rather than

$F_{\text{4xCO2,eff}}$ in Equation 2 and then substituting into Equation 6). By using instantaneous forcing, we can test the hypothesis presented in Equation 3, that the solar radiative forcing that balances a $CO_2$ quadrupling is equal to the instantaneous $CO_2$ forcing plus the sum of the radiative adjustments to the combined $CO_2$ and solar forcings. Expressed mathematically, the comparison done in Figure 11 is:

$$\left( (\Delta S_0(\%)_{\text{actual}}) \times \frac{1361 \text{ W m}^{-2}}{100\%} - 4 \times \frac{F_{\text{4xCO2,inst}}}{1-\alpha} \right) \times \frac{1-\alpha}{4} \overset{?}{=} \sum \Delta R_X \ . \qquad (7)$$

The black bars in Figure 11 show the left hand side of the Equation 7 while the gray bars show the right hand side. If the two bars are the same size, that means that the actual solar constant reduction will be equal to that predicted by Equation 3.

Instantaneous forcing, unlike effective forcing, cannot be calculated from monthly mean model output through a simple linear regression of TOA flux changes against surface temperature; instead it requires running each GCM's radiative transfer code offline with standard and quadrupled $CO_2$ concentrations. For this reason, estimates of instantaneous $CO_2$ forcing are

available for fewer models than for effective forcing. We used the "double call" instantaneous forcing estimates from the CMIP5 archive shown in Chung and Soden (2015) for the CanESM-2 and IPSL-CM5A-LR models. For three other models (CCSM4, HadGEM2-ES, and NorESM1-M), we use estimates of instantaneous $CO_2$ forcing given by Zhang and Huang (2014) based on residuals between total TOA flux changes and radiative responses calculated with radiative kernels.




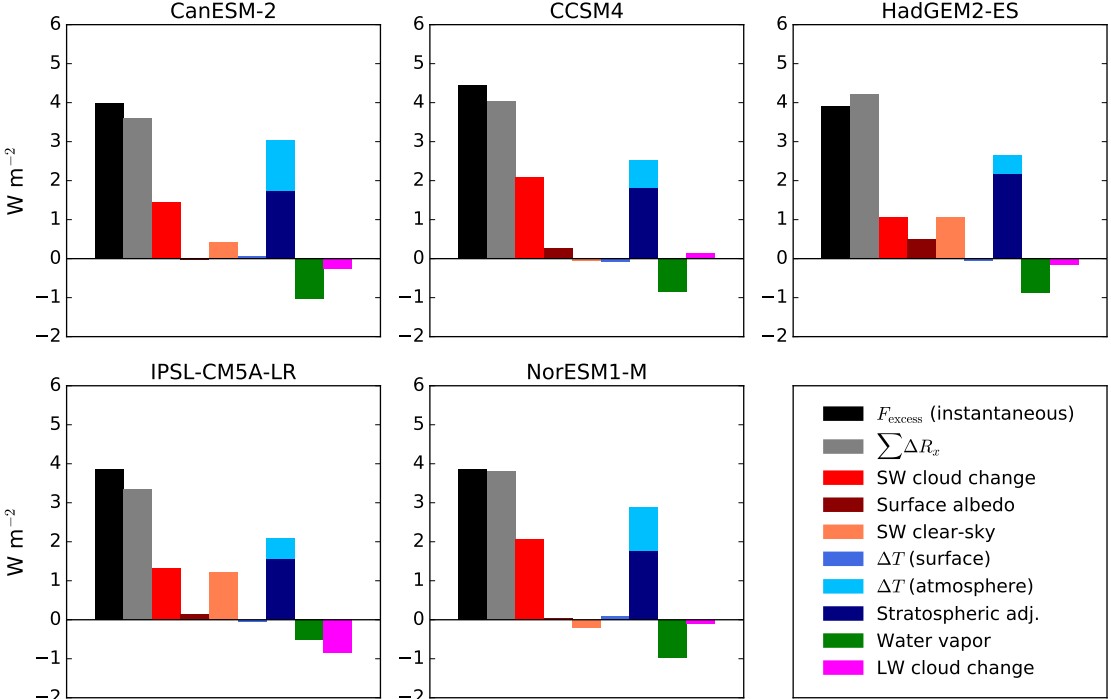

**Figure 11.** As in Figure 10 but with excess solar forcing calculated using instantaneous instead of effective $CO_2$ radiative forcing. Navy blue bar indicates the reduction in OLR due to stratospheric temperature adjustment from $CO_2$ quadrupling given by Zhang and Huang (2014), to illustrate the portion of the atmospheric temperature adjustment to G1 attributable to stratospheric cooling.

In Figure 11, the black and gray bars match to within about 10%, indicating that the theory expressed in Equation 3 works well for explaining the amount of solar constant reduction required to balance a $CO_2$ increase. The agreement is quite remarkable considering that the methods used to calculate the radiative adjustments are approximate. In evaluating the theory, it is useful to express it in terms of total instantaneous solar forcing rather than solar constant reduction:

$$F_{\text{solar,predicted}} = F_{\text{4xCO2,inst}} + \sum \Delta R_X \tag{8}$$

and compare it to the actual solar forcing in G1:

$$F_{\text{solar,actual}} = \Delta S_{0,\text{actual}}(\%) \times \frac{1361 \text{ W m}^{-2}}{100\%} \times \frac{1-\alpha}{4} \ . \tag{9}$$

These values are listed in the last two columns of Table 3. The errors in the total solar forcing in G1 predicted by Equation 8 are all within 0.5 W m$^{-2}$, or within about 6% of the total, indicating that the instantaneous solar forcing required to balance an abrupt $CO_2$ increase is well predicted by the sum of the instantaneous $CO_2$ forcing and the radiative adjustments to the combined forcings.



The two largest radiative adjustments to the G1 forcing scenario are the LW atmospheric temperature adjustment and the SW cloud adjustment. Since the temperature adjustment contains effects of both stratospheric and tropospheric temperature changes, it is worth trying to understand the partitioning between these effects. We have overlaid the OLR reduction due to the stratospheric cooling in abrupt4xCO2 given by Zhang and Huang (2014) onto the $\Delta T$ (atmosphere) bar in Figure 11. This shows that between about 50% to 75%, depending on the model, of the atmospheric temperature radiative adjustment in G1 is due to cooling of the stratosphere by the increase in $CO_2$. The rest is due to a combination of the additional cooling of the stratosphere from the reduction in insolation and the cooling of the upper troposphere which arises from the surface cooling in the tropics. The water vapor adjustment roughly compensates for the tropospheric component of the temperature adjustment, and these effects are physically linked because a cooler atmosphere emits less LW radiation but also contains less water vapor to absorb radiation from below. Therefore, the main reasons why the instantaneous solar forcing must be greater than the instantaneous $CO_2$ forcing in order to maintain energy balance are the failure to undo the stratospheric cooling and the reduction in low cloud fraction.

## 5    Conclusions

The amount of solar constant reduction required to offset an increase in $CO_2$ concentration in terms of TOA radiative balance and global mean temperature is an interesting question with implications for assessing the feasibility of solar geoengineering scenarios and for improving our theoretical understanding of the response of Earth's climate to greenhouse gas and solar forcings. This paper we explains why some intuitive predictions of the solar constant reduction are inaccurate, lays out an alternative hypothesis that the solar constant reduction can be explained based on instantaneous $CO_2$ forcings and radiative adjustments, and quantifies various radiative responses to the GeoMIP G1 scenario in order to test this hypothesis.

In the G1 experiment, at the surface, the tropics cool and the poles warm while global mean temperature remains at preindustrial. The upper troposphere experiences cooling at all latitudes, with the tropical upper troposphere cooling more than the surface. The stratosphere cools more than anywhere else in the atmosphere, due primarily to the $CO_2$ increase (Govindasamy et al., 2003). The tropospheric temperature effect is a reversal of the negative lapse rate feedback that happens in global warming simulations, in which the tropical upper troposphere warms more than the surface; in G1, because the tropics cool and the tropical temperature profile tends to follow a moist adiabat, the upper troposphere also cools, which has a warming effect on the climate by reducing OLR. Atmospheric specific humidity is reduced in the upper troposphere, which makes the atmosphere less opaque to LW radiation and largely offsets the radiative effect of the tropospheric cooling. Low cloud fraction exhibits a widespread decrease over the ocean and vegetated land areas in all models, which we attribute to decreases in boundary layer inversion strength over the ocean and reduced evaporation from plants due to the physiological response to increased $CO_2$ over land. The low cloud fraction reduction has a strong surface warming effect due to reduced reflection of sunlight by the clouds. High cloud fraction increases in the global mean in most models, but the LW radiative effect of cloud changes in G1 is slightly negative in the global, multi-model mean. When all the global mean radiative adjustments in G1 are added together, the results account, to within 10%, for the difference between the solar constant reduction that would match the instantaneous



$CO_2$ forcing and the tuned solar constant reduction that met the TOA energy balance threshold required by the G1 experiment protocol.

For future model runs of the G1 experiment, such as those being prepared for the next phase of GeoMIP corresponding to CMIP Phase 6 (Kravitz et al., 2015b), it would be useful to have a better initial guess for the solar constant reduction in order to

reduce the necessary amount of tuning. Using Equation 3 for this purpose would be tricky because the radiative responses to the combined $CO_2$ and solar forcings would be unknown before actually running the model. However, one could simply substitute an empirical value of about 4 W m$^{-2}$, a typical value for the sum of the radiative adjustments in G1 (Figure 10), for $\sum \Delta R_X$ in Equation 3. Then, tuning would only need to account for model-specific deviations from this number. If instantaneous $CO_2$ forcing was not available for a particular model, the modelers could add a correction of about 2.5 to 3 W m$^{-2}$, a typical value

for the black bars in Figure 10, to the effective $CO_2$ forcing in Equation 1.

Our analysis of the G1 experiment provides some insights into how the climate responds differently to $CO_2$ and solar forcings, but more work is necessary to better understand this question. The sums of the radiative adjustments in G1 (gray bars of Figure 11) are about 2 W m$^{-2}$ larger than the difference between effective and instantaneous forcing in abrupt4xCO2 (e.g. Table 1 of Zhang and Huang, 2014). This must be due to some combination of the solar forcing enhancing or imperfectly

canceling $CO_2$-induced radiative adjustments that warm the planet (such as the stratospheric cooling), and the solar forcing overcompensating for adjustments that cool the planet (such as the tropospheric lapse rate adjustment). Going beyond showing the stratospheric adjustment from abrupt4xCO2 in Figure 11 to separate the contributions of the $CO_2$ and solar forcings to the radiative adjustments in G1 would be nontrivial. Regressing the APRP- and kernel-derived radiative responses in the abrupt4xCO2 experiment against global mean temperature change to obtain the rapid adjustments to the $CO_2$ quadrupling

would run into issues with accuracy due to nonlinearity of feedbacks with temperature increases that would skew the location of the intercept (Armour et al., 2013), so an analysis of GCM runs with increased $CO_2$ and fixed SSTs would be necessary. Furthermore, it may not be the case that the rapid adjustments to the two forcings add together linearly. While some variables, such as global mean temperature, respond linearly to different combinations of $CO_2$ and solar forcings (Kravitz et al., 2015a), other aspects of the climate system are inherently nonlinear. LW emission goes with the fourth power of temperature, and spe-

cific humidity rises exponentially with temperature, a relationship that affects atmospheric energy transport and the meridional temperature gradient (Hwang et al., 2011; Russotto and Ackerman, 2018). The interactions between the exponential dependence of specific humidity and the 4th power dependence of LW emission on temperature may affect the extent to which the water vapor and tropospheric temperature adjustments compensate for each other, as they seem to roughly cancel in G1 but the water vapor feedback exceeds the lapse rate feedback in global warming simulations (Soden and Held, 2006; Soden et al.,

2008; Vial et al., 2013). The water vapor and lapse rate adjustments are dependent on the pattern of tropical cooling and polar warming which might not occur if a latitudinal distribution of solar reflection was targeted to cool the poles more (Ban-Weiss and Caldeira, 2010; Kravitz et al., 2016).

It would be very interesting to study how cloud rapid adjustments and feedbacks differ under solar versus $CO_2$ forcing in a model intercomparison framework. The cloud fraction changes in G1 imply that rapid cloud responses to $CO_2$ and solar

forcings are different, but this requires further study with GCM runs that perturb only the solar constant and not $CO_2$. Since the





global mean temperature does not change, the G1 experiment tells us very little about cloud feedbacks, which are temperature dependent. An attempt was made (Huneeus et al., 2014) to study cloud rapid adjustments and feedbacks under solar forcings by subtracting the G1 experiment from the abrupt4xCO2 experiment, but this approach is bound to produce similar feedback parameters for this "solar" forcing versus the abrupt4xCO2 - piControl $CO_2$ forcing because, while there are two different

baselines, there is only one perturbation run, abrupt4xCO2, that has a global mean temperature change onto which radiative flux changes can be regressed. Some studies have included solar-only GCM runs (e.g. Bala et al., 2008; Schaller et al., 2013, 2014; Modak et al., 2016), but these have included only one or two models, and while some, such as Modak et al. (2016), have looked at cloud radiative effects and cloud fraction, none have used methods that account for cloud masking to isolate the radiative effects of physical cloud changes. There is no solar equivalent of abrupt4xCO2 in CMIP5 or any of its associated

projects; the closest analogue is probably the aerosol-forcing-only historical runs from the CMIP5 "historicalMisc" collection, analyzed, *e.g.*, by Salzmann (2016). The Precipitation Driver and Response Model Intercomparison Project (Myhre et al., 2017) includes a solar constant increase experiment and may be a good avenue to explore cloud changes under solar forcings.

    If we were thinking about actually doing solar geoengineering, using Equation 3 to predict the necessary solar reflection would be hampered by the fact that we would not know the radiative responses to the intervention *a priori*. Estimates of these

adjustments from models would be subject to uncertainty (note the inter-model spread of 2 W m$^{-2}$ in the gray bars of Figure 10), and various aspects of the current anthropogenic radiative forcing, particularly aerosol forcing, also have large uncertainty (Myhre et al., 2013). Observing the climate response to a smaller-scale geoengineering test that would impose a measurable change in the global mean radiation balance (e.g. Keith et al., 2014) could provide a better estimate of these quantities. Such a test would pose ethical questions related to justice, compensation and informed consent similar to those for a full deployment

(Lenferna et al., 2017). Another option would be to actively control the global mean temperature by adjusting the amount of solar reflection every year in response to observations (Kravitz et al., 2014). If solar geoengineering was attempting to actually cool the planet instead of simply preventing future warming under increasing $CO_2$, then temperature-dependent feedbacks on the solar forcing, which are not captured by the G1 experiment, would affect the amount of solar geoengineering required, as would time-dependent effects from the inertia of ocean temperature changes. Analysis of other GeoMIP experiments, such

as G4, that do impose a global mean temperature change from the solar forcing, could help illustrate these issues. If solar geoengineering was to be done using stratospheric aerosols, then an additional layer of uncertainty regarding microphysical and chemical effects would impact the amount of aerosol injection required to achieve the desired forcing, as summarized by Visioni et al. (2017).

    Besides their effects on the required solar forcing, the changes in atmospheric physical properties that occur in G1 are

interesting in their own right, and may have policy implications if they translated to a real geoengineering deployment. If low cloud fraction were actually reduced by solar geoengineering, it could result in increased solar energy production, and could enhance vegetation growth in sunlight-limited regimes like the Amazon (Nemani et al., 2003). On the other hand, a reduction in low clouds over the ocean would make it more difficult to do marine cloud brightening at the same time as other forms of solar geoengineering. Changes in cirrus clouds are also relevant in the context of research on the effects of sedimentation of

injected stratospheric aerosols on high clouds (Kuebbeler et al., 2012; Visioni et al., 2018) and proposals to intentionally thin



cirrus clouds with nucleation-inducing aerosols in order to cool the earth by increased LW emission (Mitchell and Finnegan, 2009). The increase in high clouds in most models in G1 indicates that thermodynamic and radiative adjustments to the forcing scenario can have effects on high clouds that may counteract unintentional or intentional microphysical effects. Our analysis of G1 also illustrates that stratospheric ozone could be affected by changes in stratospheric water vapor resulting from TTL

temperature changes. In model runs with actual injection of sulfate aerosols, LW absorption of these particles warms the tropical tropopause and increases stratospheric water vapor, which results in decreased ozone concentrations (Heckendorn et al., 2009). Keith et al. (2016) suggest that this risk could be mitigated by instead injecting calcite aerosols, which would absorb much less LW radiation than sulfates, but the inconsistency between models in stratospheric water vapor responses to the G1 experiment, which includes no aerosol injection in G1, shows that much uncertainty remains in this area. Taken together, these issues

emphasize the importance of continuing to perform and analyze geoengineering simulations, both in highly idealized scenarios like G1 and more realistic ones like G4 or G4SSA (Tilmes et al., 2015), in order to better understand the climate responses to geoengineering schemes and the different roles played by thermodynamics, radiation, microphysics and chemistry in these responses.

*Author contributions.* R.D. Russotto analyzed the GCM output, produced the figures, and wrote the bulk of the paper. T.P. Ackerman pro-

vided general guidance and assisted with the preparation of the manuscript.

*Code availability.* All scripts used to analyze data and create plots are available here:
https://atmos.washington.edu/~russotto/G1_clouds_s0_paper_scripts/index.html
They will be posted to a permanent repository upon acceptance of the paper.

*Acknowledgements.* This work was supported by a grant to JISAO from the Fund for Innovative Climate and Energy Research. For their

roles in producing, coordinating, and making available the CMIP5 and GeoMIP model output, we acknowledge the climate modeling groups (listed in Table 1 of this paper), the World Climate Research Programme's (WCRP) Working Group on Coupled Modelling (WGCM), and the Global Organization for Earth System Science Portals (GO-ESSP). We are grateful to Duoying Ji, Ben Kravitz, Helene Muri, Ulrike Niemeier, Stephen Phipps, and Jin-Ho Yoon for helping to provide access to GeoMIP output that was not available through online repositories; to Karen Shell for helping to explain how to use the Shell et al. (2008) radiative kernels; and to Eui-Seok Chung for providing numbers for the double-

call instantaneous forcing plotted in Chung and Soden (2015). We thank Dargan Frierson, Blaž Gasparini, Cristian Proistosescu, and Brian Rose for discussions that helped to influence the paper.



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
