# Peer review of "Changes in clouds and thermodynamics under solar geoengineering and implications for required solar reduction"

_Atmospheric Chemistry and Physics, 2018_

## Referee Comment (RC1) · Anonymous Referee #1 · 25 May 2018

**Review of the manuscript "Changes in clouds and thermodynamics under solar geoengineering and implications for required solar reduction" submitted by R. D. Russotto and T. P. Ackerman for publication in ACP**

In the G1 simulation of GeoMIP it is attempted to reach TOA radiative balance under instantaneous quadrupling of CO2 by a decrease of solar irradiance. It had been recognized earlier that the solar constant has to be reduced more than estimated using simple assumptions, i.e. a change of solar irradiance seems less effective than the changing CO2, but to my knowledge, the reasons for this behavior had not been comprehensively analyzed. This paper closes this gap in a, in my opinion, very convincing way, at least to the extent this is possible with the available model output. The authors have used direct analysis of model output and the technique of radiative kernels to infer that the LW adjustment due to changes in atmospheric (in particular stratospheric) temperatures and the SW adjustment related to decreasing low cloud cover in G1 are the largest contributors to the underestimation of the necessary solar constant reduction. I think these results are very useful for the discussion of solar geoengineering methods but also for understanding responses to solar and GHG forcing. I find the paper in general very well written, and the selection and quality of tables and figures very appropriate. In summary I recommend publication of this paper after consideration of the following minor issues:

- P5L20: "ant"
- In several places, the overall cooling in the free troposphere is mentioned, which is explained with the surface cooling and the temperature profile following a moist adiabat (page 6). However, the globally averaged surface temperature does not increase in G1. It would be useful to mention why the tropical cooling dominates the temperature change in the free troposphere.
- P9L18: It is argued that a reduction in low clouds would partially counteract the cooling from solar geoengineering. However, the authors do are not able to discriminate if CO2 or solar irradiance changes lead to the adjustments. Hence, I don't understand why this change in clouds is related to the irradiance change.
- Table 3: It might be useful to add a line with the multi-model mean (which is even referred to in the text) to this table, and possibly also Table 2.
- At some places (e.g. P22L8) the authors talk about something "predicted" by Eq. 8. I think it would be useful to mention already early (it is done only in the Conclusions) that this is a prediction in hindsight, based on the analysis of the model results, or even avoid the word "prediction".
- Last paragraph of Section 4. I'm wondering if this shouldn't go to Section 3.
- P25L17: I find the remark on a "smaller-scale geoengineering test" not specific enough. What means smaller? I think also the reference is unfortunate, because Keith et al. (2014) only cite another paper (McMynowski et al., 2011) for providing an estimate of what small could mean and I have the impression they cite them wrongly. So I'd suggest to rather cite the original publication and be a bit more specific.
- P25L21: "If solar geoengineering was attempting to cool the planet instead of simply preventing future warming … then temperature dependent feedbacks …" I don't understand this statement. It all depends on the reference. If it is preindustrial climate then also a stabilization would include temperature feedbacks. And in the case of a cooling these would depend on the amount of cooling.

---

## Referee Comment (RC2) · Anonymous Referee #2 · 1 Jun 2018

Review of "Changes in clouds and thermodynamics under solar geoengineering and implications for required solar reduction" by Russotto and Ackerman, submitted to Atmospheric Chemistry and Physics

This manuscript presents multi-model analysis of simulations from the GEOMIP G1 experiment, in which $CO_2$ is quadrupled while the solar constant has been reduced sufficiently (through iteration) to achieve near zero global mean surface temperature change. The work is motivated by the observation that the required solar constant reduction is greater than the value that would exactly offset the effective radiative forcing from $4xCO_2$ in the global mean planetary energy budget. Well-established tools

such as radiative kernels and the APRP method are used to quantify the partial contributions of various cloud and clear-sky mechanisms to the total radiative changes in order to understand why a greater than expected solar constant reduction is required. A key finding is that there is a widespread, robust reduction in low cloud fraction in the models, which increases the necessary solar constant reduction to offset $CO_2$-driven warming. This cloud reduction is at least qualitatively consistent with widespread reductions in two different measures of lower tropospheric stability.

This manuscript was well-written (although I did find it rather long and wordy at times). The figures are good and I find no fault with any of the analyses. My main difficulty with this work is with the framing of the central question and many of the results. Equation (3) is just a statement of an energy budget. The authors state (page 5, line 15) that they will "test the hypothesis that the solar constant reduction can be predicted using Equation 3". I don't think there is any such hypothesis, because Equation (3) has no predictive power until the adjustment terms are known. And the sum of the adjustment terms are by definition what's needed to close the budget. So I see circular reasoning. The only way to use this framework to get \Delta S_0 is to calculate the adjustments, but this is only done a posteriori by running the models. This is in fact noted by the authors several times (page 24, line5; page 25, line 13). Given this limitation, I don't see why the authors are presenting this work as a test of such a hypothesis. The fact that the budgets approximately balance in Figure 11 is really just an approximate validation of the analysis techniques (kernels, APRP) – it does not represent a conceptual validation of any physical or predictive framework.

To me a more interesting question would be to look at differences between adjustments *actually achieved by the models* (as analyzed here) and the traditional notion of *adjusted radiative forcings* for which SSTs are held fixed. These will not be the same in this context, because even though the simulations feature near-zero surface warming, there are *local* SST changes almost everywhere, which surely have interesting consequences for atmospheric stability and cloud processes. There may be (probably

are) "rapid adjustments" to solar forcing that are quite different than the eventual comprehensive adjustment of the models after allowing the SSTs to change. These could be evaluated by carrying out Hansen-style fixed SST experiments, even with a single model. The manuscript does not say much about the role of the flattened equator-to-pole temperature gradient on the radiative effects, which seems like a missed opportunity to learn more about the relevant physics.

That said, I think the results themselves are interesting and sound, and they should be published after a slight reframing of the central questions. Keep the focus simply on answering why the required solar constant reduction is larger than expected.

My other substantial criticism is regarding timescales. Nowhere in this manuscript did I see any mention of the transient nature of the response. This seems important enough to merit some thought. As far as I've understood, these are coupled model simulations. The climate will continue to adjust long after 50 years, with implications for the spatial pattern of SSTs and consequent radiative feedback processes. The paper seems to treat the 50-year response as an equilibrium, which it surely is not. If I have misunderstood and these are actually slab ocean calculation, then the interpretation is more appropriate, but this should be clarified in the text.

Detailed issues:

- Page 2, Line 25: "One might intuitively expect...." This seems like a strawman argument. I would not expect this. Forcing and feedback are not the same thing. If others have suggested that these things should be correlated, then provide a citation here.

- Page 5, line 1: A reference to Hansen et al. (2005) would be helpful for readers who need clarification about the various concepts of radiative forcing.

- Page 5, line 11: It was not clear at first why the authors are referring to 50 years here. Later it becomes evident that that is the length of the GEOMIP simulations. That should be clarified.

[Figure]

- Page 5, line 20: "ant"

- Figure 6: the results here are presented qualitatively. Why not compute a spatial correlation between EIS and low cloud changes?

- Page 13, first paragraph: This is a good discussion of cloud controlling factors.

- Page 13, bottom: I realize that these are more speculations than results per se, but I feel like this discussion confuses stocks vs. flows of water vapor. Reduced evaporation does not necessarily imply reduced boundary layer humidity.

- Page 17, line 14: I find it convoluted to describe the decreased OLR due to cooler temperatures to be "warming effect"

- Page 17, line 19: I think these statements are inaccurate. The correction is not just for cloud masking of $CO_2$ changes. A more important correction embedded in equation (5) is that differences in CRE depend on clear sky changes as well as cloud changes.

- Figure 10 and 11: I guess I'm not sure why these are presented as separate figures? The authors are quick to point out that figure 10 is misleading (page 21, line 15). Why not combine Figs. 10 and 11 and avoid potential confusion.

- Page 23, line 5: This is the traditional "stratosphere adjusted" contribution to radiative forcing, e.g. Hansen et al. (2005)

References

J. Hansen, M. Sato, R. Ruedy, L. Nazarenko, A. Lacis, G. A. Schmidt, G. Russell, I. Aleinov, M. Bauer, S. Bauer, N. Bell, B. Cairns, V. Canuto, M. Chandler, Y. Cheng, A. D. Genio, G. Faluvegi, E. Fleming, A. Friend, T. Hall, C. Jackman, M. Kelley, N. Kiang, D. Koch, J. Lean, J. Lerner, K. Lo, S. Menon, R. Miller, P. Minnis, T. Novakov, V. Oinas, J. Perlwitz, J. Perlwitz, D. Rind, A. Romanou, D. Shindell, P. Stone, S. Sun, N. Tausnev, D. Thresher, B. Wielicki, T. Wong, M. Yao, and S. Zhang. Efficacy of climate forcings. J. Geophys. Res., 110(D18104), 2005.

---

## Referee Comment (RC3) · Anonymous Referee #3 · 7 Jun 2018

General comments:

The manuscript analyses results from experiment G1 under the model intercomparison on geoengineering (GeoMIP). Here, the solar constant is reduced so as to keep the top of the atmosphere radiative balance at zero when abruptly quadrupling the atmospheric $CO_2$ concentration. The article discusses why the reduction in solar constant must be larger than what is predicted based on instantaneous and long-term effects of the $CO_2$ increase alone and that rapid adjustments to the change in solar radiation must be taken into account. The manuscript is very well written and easy to follow, the authors present good graphics and tables to support their discussion and they use suited sci-

entific methods in form of radiative kernals to decompose their radiative responses into rapid adjustments of different parts of the climate system. My only major objection is how the discussion is angled towards the potential use of their Equation 3, which to me is just a description of the radiative processes involved. The terms of this equation cannot be known before the experiment is run and it is therefore a bit misleading to suggest that it could be used to find a suited change in solar constant to counteract some positive forcing on the climate system. The equation does, however, support the results from the radiative kernels and should rather be used as a tool to test the validity of these results.

Specific comments

- P1, L21: "High cloud fraction increases.." Intuitively, this should also give a warming effect. You show later that this effect is not. I suggest to mention this here or to rewrite a bit. I was a bit puzzled why the increases in high clouds were not part of the following sentence on warming effects.

- P3, eq (2): Though obvious to most, perhaps you should state where this number comes from?

- P5, L8. "Existing tools. . ." A bit vague. Mention kernels already here.

- P5, L10: ". . .and not "feedbacks". . .". This is true on a global scale, but is it true on a regional scale? You state that the surface temperature increase in the polar regions and Figure 7c shows a clear effect on solar radiation from surface albedo changes.

- P5, L20: ". . .ant . . ."

- P9, L18: "An increase in high cloud fraction would. . ." This is true and in this experiments, the temperature at these altitudes have decreased, which should enhance the effect further?

- P9, L22: ". . .due to a reduction in LW emission from the cloud top" I am confused – how does something get warmer from receiving less radiation? Please clarify this

sentence. Reduced high cloud cover will expose the region above to more surface radiation as this is now allowed to reach higher altitudes.

- P16, table 3: Can you not just put a minus sign in front of the solar forcing to avoid the +/- confusion?

- P19,L1: "...LW effect of the decrease in low cloud cover fraction compensates for this." Is the temperature difference between the cloud top and the surface large enough to explain this? Or are you discussing surface warming here? Please clarify.

- Conclusions: I find the conclusion of the manuscript a bit wordy and I suggest trimming this section and making more about the results in this study rather than a speculation into future possible geoengineering challenges.

- P23, L17: Typo "This paper we explains.."

- P25, L12: The PDRMIP experiments actually contains multimodel runs of both reduced solar constant and increased $CO_2$ and could therefor provide the data you need to look further into this.

- P25, L21: "If solar geoengineering was attempting to actually cool the planet..." What? Back to preindustrial conditions?

---

## Author Comment (AC1) · 24 Jul 2018

Responses to Reviewer Comments on "Changes in clouds and thermodynamics under solar geoengineering and implications for required solar reduction", by Rick Russotto and Thomas Ackerman

Responses are in blue below individual comments. Appended to this document are a redlined manuscript showing the changes, a clean revised manuscript, and a revised version of the Supporting Information. In addition to the changes in response to these comments, some minor edits have been made to the discussion of LTS and EIS changes in response to Rob Wood's comments on Rick Russotto's doctoral dissertation.

**Reviewer 1**

In the G1 simulation of GeoMIP it is attempted to reach TOA radiative balance under instantaneous quadrupling of CO2 by a decrease of solar irradiance. It had been recognized earlier that the solar constant has to be reduced more than estimated using simple assumptions, i.e. a change of solar irradiance seems less effective than the changing CO2, but to my knowledge, the reasons for this behavior had not been comprehensively analyzed. This paper closes this gap in a, in my opinion, very convincing way, at least to the extent this is possible with the available model output. The authors have used direct analysis of model output and the technique of radiative kernels to infer that the LW adjustment due to changes in atmospheric (in particular stratospheric) temperatures and the SW adjustment related to decreasing low cloud cover in G1 are the largest contributors to the underestimation of the necessary solar constant reduction. I think these results are very useful for the discussion of solar geoengineering methods but also for understanding responses to solar and GHG forcing. I find the paper in general very well written, and the selection and quality of tables and figures very appropriate.

In summary I recommend publication of this paper after consideration of the following minor issues:

• P5L20: "ant"

This typo has been corrected in the revised version.

• In several places, the overall cooling in the free troposphere is mentioned, which is explained with the surface cooling and the temperature profile following a moist adiabat (page 6). However, the globally averaged surface temperature does not increase in G1. It would be useful to mention why the tropical cooling dominates the temperature change in the free troposphere.

It is difficult to find discussion of this issue in existing papers (like Govindasamy et al. 2003), and it is not immediately obvious what the explanation is. It could be that the cooling throughout the free troposphere is due to poleward advection of the cooled tropical upper tropospheric air by the Hadley circulation, or that less water vapor is available to heat the upper atmosphere through latent heating arising from local deep convection. Absent time to thoroughly explore this issue, we do not explicitly address it.

• P9L18: It is argued that a reduction in low clouds would partially counteract the cooling from solar geoengineering. However, the authors do are not able to discriminate if CO2 or solar irradiance changes lead to the adjustments. Hence, I don't understand why this change in clouds is related to the irradiance change.

The word "counteract" has been replaced with "offset", which seems less likely to imply that the clouds are definitely caused by the sunlight reduction. (We don't explicitly state this anywhere, and say in the conclusions that we don't know the relative contributions of the solar vs. CO2 forcings to the adjustments.)

• Table 3: It might be useful to add a line with the multi-model mean (which is even referred to in the text) to this table, and possibly also Table 2.

Multi-model means have been added to Tables 2 and 3.

• At some places (e.g. P22L8) the authors talk about something "predicted" by Eq. 8. I think it would be useful to mention already early (it is done only in the Conclusions) that this is a prediction in hindsight, based on the analysis of the model results, or even avoid the word "prediction".

See response to Reviewer 2's first major comment.

• Last paragraph of Section 4. I'm wondering if this shouldn't go to Section 3.

We have left this paragraph where it is because it follows the first introduction of the instantaneous forcing estimates.

• P25L17: I find the remark on a "smaller-scale geoengineering test" not specific enough. What means smaller? I think also the reference is unfortunate, because Keith et al. (2014) only cite another paper (McMynowski et al., 2011) for providing an estimate of what small could mean and I have the impression they cite them wrongly. So I'd suggest to rather cite the original publication and be a bit more specific.

We have added a reference to the MacMynowski et al. paper and clarified what is meant by smaller scale (about $1/10^{th}$ the radiative forcing relative to deployment, and lasting about a decade).

• P25L21: "If solar geoengineering was attempting to cool the planet instead of simply preventing future warming … then temperature dependent feedbacks …" I don't understand this statement. It all depends on the reference. If it is preindustrial climate then also a stabilization would include temperature feedbacks. And in the case of a cooling these would depend on the amount of cooling.

We have expanded on this discussion a bit to provide examples of what a cooling scenario might look like, and to acknowledge this point but also emphasize that we would not be in an equilibrium situation and the operation of the feedbacks would be affected by the inertia associated with ocean heat storage.

**Reviewer 2**

This manuscript presents multi-model analysis of simulations from the GEOMIP G1 experiment, in which CO2 is quadrupled while the solar constant has been reduced sufficiently (through iteration) to achieve near zero global mean surface temperature change. The work is motivated by the observation that the required solar constant reduction is greater than the value that would exactly offset the effective radiative forcing from 4xCO2 in the global mean planetary energy budget. Well-established tools such as radiative kernels and the APRP method are used to quantify the partial contributions of various cloud and clear-sky mechanisms to the total radiative changes in order to understand why a greater than expected solar constant reduction is required. A key finding is that there is a widespread, robust reduction in low cloud fraction in the models, which increases the necessary solar constant reduction to offset CO2-driven

warming. This cloud reduction is at least qualitatively consistent with widespread reductions in two different measures of lower tropospheric stability.

This manuscript was well-written (although I did find it rather long and wordy at times). The figures are good and I find no fault with any of the analyses. My main difficulty with this work is with the framing of the central question and many of the results. Equation (3) is just a statement of an energy budget. The authors state (page 5, line 15) that they will "test the hypothesis that the solar constant reduction can be predicted using Equation 3". I don't think there is any such hypothesis, because Equation (3) has no predictive power until the adjustment terms are known. And the sum of the adjustment terms are by definition what's needed to close the budget. So I see circular reasoning. The only way to use this framework to get \Delta S_0 is to calculate the adjustments, but this is only done *a posteriori* by running the models. This is in fact noted by the authors several times (page 24, line5; page 25, line 13). Given this limitation, I don't see why the authors are presenting this work as a test of such a hypothesis. The fact that the budgets approximately balance in Figure 11 is really just an approximate validation of the analysis techniques (kernels, APRP) – it does not represent a conceptual validation of any physical or predictive framework.

These criticisms of the framing of the results are valid. Relevant portions of the manuscript have been edited so that Equation 3 is not presented as a hypothesis to be tested or having predictive value, and so that characterizing the radiative adjustments and explaining the required solar constant reduction is presented as the main focus of the paper. For Equations 1 and 2, however, we have continued referring to "prediction", since it makes sense in the context of the initial guess for the solar constant reduction used by the modeling groups.

To me a more interesting question would be to look at differences between adjustments *actually achieved by the models* (as analyzed here) and the traditional notion of *adjusted radiative forcings* for which SSTs are held fixed. These will not be the same in this context, because even though the simulations feature near-zero surface warming, there are *local* SST changes almost everywhere, which surely have interesting consequences for atmospheric stability and cloud processes. There may be (probably are) "rapid adjustments" to solar forcing that are quite different than the eventual comprehensive adjustment of the models after allowing the SSTs to change. These could be evaluated by carrying out Hansen-style fixed SST experiments, even with a single model. The manuscript does not say much about the role of the flattened equator-to-pole temperature gradient on the radiative effects, which seems like a missed opportunity to learn more about the relevant physics.

Doing fixed-SST experiments of the response to solar forcing and the G1 scenario is a good idea, but outside the scope of this paper.

That said, I think the results themselves are interesting and sound, and they should be published after a slight reframing of the central questions. Keep the focus simply on answering why the required solar constant reduction is larger than expected.

My other substantial criticism is regarding timescales. Nowhere in this manuscript did I see any mention of the transient nature of the response. This seems important enough to merit some thought. As far as I've understood, these are coupled model simulations. The climate will continue to adjust long after 50 years, with implications for the spatial pattern of SSTs and consequent radiative feedback processes. The paper seems to treat the 50-year response as an equilibrium, which it surely is not. If I have misunderstood and these are actually slab ocean calculation, then the interpretation is more appropriate, but this should be clarified in the text.

These are fully coupled runs, but we don't see any reason to expect drift given previously published time series plots of the surface energy budget (Kravitz et al. 2013b, cited in the paper), and found none in time series plots of the APRP responses that are now included as Figure S13. We now discuss the time scale of averaging and the lack of drift in the first paragraph of Section 2.

Detailed issues:

- Page 2, Line 25: "One might intuitively expect...." This seems like a strawman argument. I would not expect this. Forcing and feedback are not the same thing. If others have suggested that these things should be correlated, then provide a citation here.

This argument was included because some people inside and outside our group (who will remain unnamed for privacy reasons) were surprised by the lack of a correlation in Figure 1a. However, on looking through the literature for previous comments on this issue, it appears that it has already been acknowledged that ECS should not affect the required amount of solar geoengineering in the case of no global mean temperature change, going back to Matthews and Caldeira (2007) who demonstrated this in more idealized simulations than those in GeoMIP. We now cite that paper in the manuscript, and have revised the discussion of the ECS vs. solar constant reduction scatter plot to focus on ruling out feedbacks as a cause of inter-model spread in the solar constant reduction, rather than disproving this theory. Also, to improve the flow of the manuscript given this change, we have flipped the order of the panels in Figure 1, and discuss the formula based on effective $CO_2$ forcing first.

- Page 5, line 1: A reference to Hansen et al. (2005) would be helpful for readers who need clarification about the various concepts of radiative forcing.

The suggested reference has been added.

- Page 5, line 11: It was not clear at first why the authors are referring to 50 years here. Later it becomes evident that that is the length of the GEOMIP simulations. That should be clarified.

We have moved the first reference to the 50 year time scale to the beginning of Section 2, when we explain the averaging period and the fact that the models were only run out for 50 years for G1.

- Page 5, line 20: "ant"

This typo has been corrected in the revised version.

- Figure 6: the results here are presented qualitatively. Why not compute a spatial correlation between EIS and low cloud changes?

This suggestion is beyond the scope of the paper. This type of analysis would be nontrivial, partly due to there being a number of possible confounding factors. For example, the climatological correlation between EIS and low clouds may vary between different locations based on local conditions.

- Page 13, first paragraph: This is a good discussion of cloud controlling factors.

Thanks. It is still in the paper.

- Page 13, bottom: I realize that these are more speculations than results per se, but I feel like this discussion confuses stocks vs. flows of water vapor. Reduced evaporation does not necessarily imply reduced boundary layer humidity.

While it is true that this linkage is not necessarily true in general, it makes more sense over land where moisture is limited than for the planet as a whole. The cited Cao *et al.* reference states: "This decrease in relative humidity is associated with the diminished source of water vapor to the atmosphere as a result of decreased canopy transpiration."

- Page 17, line 14: I find it convoluted to describe the decreased OLR due to cooler temperatures to be "warming effect"

We now simply refer to OLR changes in this sentence.

- Page 17, line 19: I think these statements are inaccurate. The correction is not just for cloud masking of CO2 changes. A more important correction embedded in equation (5) is that differences in CRE depend on clear sky changes as well as cloud changes.

This sentence has been edited to clarify what is meant by cloud masking.

- Figure 10 and 11: I guess I'm not sure why these are presented as separate figures? The authors are quick to point out that figure 10 is misleading (page 21, line 15). Why not combine Figs. 10 and 11 and avoid potential confusion.

The choice of presenting Figures 10 and 11 separately allows for the results for the models that lacked  to be presented in Figure 10, while consistently showing the same information across each panel of each figure. The black bars in Figure 10 are also relevant to the

- Page 23, line 5: This is the traditional "stratosphere adjusted" contribution to radiative forcing, e.g. Hansen et al. (2005)

This is now noted in the paper, with the suggested reference.

**Reviewer 3**

The manuscript analyses results from experiment G1 under the model intercomparison on geoengineering (GeoMIP). Here, the solar constant is reduced so as to keep the top of the atmosphere radiative balance at zero when abruptly quadrupling the atmospheric CO2 concentration. The article discusses why the reduction in solar constant must be larger than what is predicted based on instantaneous and long-term effects of the CO2 increase alone and that rapid adjustments to the change in solar radiation must be taken into account. The manuscript is very well written and easy to follow, the authors present good graphics and tables to support their discussion and they use suited scientific methods in form of radiative kernals to decompose their radiative responses into rapid adjustments of different parts of the climate system. My only major objection is how the discussion is angled towards the potential use of their Equation 3, which to me is just a description of the radiative processes involved. The terms of this equation cannot be known before the experiment is run and it is therefore a bit misleading to suggest that it could be used to find a suited change in solar constant to counteract some positive forcing on the climate system. The equation does, however, support the results from the radiative kernals and should rather be used as a tool to test the validity of these results.

On the paper's framing and discussion of Equation 3, see response to Reviewer 2's first major comment.

Specific comments

- P1, L21: "High cloud fraction increases.." Intuitively, this should also give a warming effect. You show later that this effect is not. I suggest to mention this here or to rewrite a bit. I was a bit puzzled why the increases in high clouds were not part of the following sentence on warming effects.

We now mention the small LW cloud adjustment in the abstract.

- P3, eq (2): Though obvious to most, perhaps you should state where this number comes from?

We now write S0 in the equation and list and explain the 1361 number in the text.

- P5, L8. "Existing tools. . ." A bit vague. Mention kernals already here.

We now mention APRP and radiative kernels at the first mention of existing analysis tools, and cite the relevant papers.

- P5, L10: ". . .and not "feedbacks". . .". This is true on a global scale, but is it true on a regional scale? You state that the surface temperature increase in the polar regions and Figure 7c shows a clear effect on solar radiation from surface albedo changes.

We now note that local temperature changes are responsible for many of these radiative adjustments. We have also changed "it is appropriate to refer to…" to "we refer to…" to acknowledge the possibility that others might choose to define adjustments and feedbacks differently.

- P5, L20: ". . .ant . . ."

This typo has been corrected in the revised version.

- P9, L18: "An increase in high cloud fraction would. . ." This is true and in this experiments, the temperature at these altitudes have decreased, which should enhance the effect further?

The enhancement of CRE by the lower temperatures for pre-existing clouds should be accounted for by the correction in Eq. 5, but for the new clouds this effect will show up in Figure 9. However, since the overall LW cloud adjustment in G1 is small, it does not seem worth it to mention this enhancing effect in the text.

- P9, L22: ". . .due to a reduction in LW emission from the cloud top" I am confused – how does something get warmer from receiving less radiation? Please clarify this sentence. Reduced high cloud cover will expose the region above to more surface radiation as this is now allowed to reach higher altitudes.

The physical argument is that the top of the cloud is being deprived of a radiative heat sink and therefore the region around the cloud top gets warmer. However, we recognize that this could be confusing and it is not important to the overall narrative, so we have taken this sentence out.

- P16, table 3: Can you not just put a minus sign in front of the solar forcing to avoid the +/- confusion?

We said early on that we were dropping the negative on the solar constant change, so we also do this for the solar forcings for internal consistency.

- P19,L1: *. . .LW effect of the decrease in low cloud cover fraction compensates for this." Is the temperature difference between the cloud top and the surface large enough to explain this? Or are you discussing surface warming here? Please clarify.

We meant the former, and have added a clarification here. The idea is that although the low clouds have a smaller temperature difference relative to the surface, the low cloud changes occur over a wide area. The effect of the surface temperature change is accounted for in the surface temperature kernel figure, and the masking of this effect by the pre-existing clouds is accounted for by the correction in Equation 5.

- Conclusions: I find the conclusion of the manuscript a bit wordy and I suggest trimming this section and making more about the results in this study rather than a speculation into future possible geoengineering challenges.

While the conclusions have been edited in response to other comments, we have not removed the speculation about future geoengineering challenges because we think the geoengineering research community will be interested in this discussion.

- P23, L17: Typo "This paper we explains.."

This typo has been corrected in the revised version.

- P25, L12: The PDRMIP experiments actually contains multimodel runs of both reduced solar constant and increased CO2 and could therefor provide the data you need to look further into this.

According to Table 1 of the cited PDRMIP paper, they have a CO2 doubling run but not a solar constant decrease run—only a solar constant increase. However, it turns out both abrupt solar increase and decrease experiments are included as part of CMIP6 (specifically the CFMIP component); a reference for this is now cited.

- P25, L21: "If solar geoengineering was attempting to actually cool the planet. . ." What? Back to preindustrial conditions?

See response to Reviewer 1's comment on this sentence. We have added examples for possible goals of cooling.

[revised manuscript text omitted]

---

## Author Response (AR2)

No specific revisions were requested by the editor, so the same manuscript and supplement files from the first revision have been re-uploaded.